# Steering When Necessary: Flexible Steering Large Language Models with Backtracking

**Zifeng Cheng**[*], **Jinwei Gan**[*], **Zhiwei Jiang**[†],
**Cong Wang**, **Yafeng Yin**, **Xiang Luo**, **Yuchen Fu**, **Qing Gu**
State Key Laboratory for Novel Software Technology, Nanjing University, China
`chengzf@nju.edu.cn, ganjw@smail.nju.edu.cn, jzw@nju.edu.cn,`
`cw@smail.nju.edu.cn, yafeng@nju.edu.cn,`
`{luoxiang,yuchenfu}@smail.nju.edu.cn, guq@.nju.edu.cn`

## Abstract

Large language models (LLMs) have achieved remarkable performance across many generation tasks. Nevertheless, effectively aligning them with desired behaviors remains a significant challenge. Activation steering is an effective and cost-efficient approach that directly modifies the activations of LLMs during the inference stage, aligning their responses with the desired behaviors and avoiding the high cost of fine-tuning. Existing methods typically indiscriminately intervene to all generations or rely solely on the question to determine intervention, which limits the accurate assessment of the intervention strength. To this end, we propose the **F**lexible **A**ctivation **S**teering with **B**acktracking (**FASB**) framework, which dynamically determines both the necessity and strength of intervention by tracking the internal states of the LLMs during generation, considering both the question and the generated content. Since intervening after detecting a deviation from the desired behavior is often too late, we further propose the backtracking mechanism to correct the deviated tokens and steer the LLMs toward the desired behavior. Extensive experiments on the TruthfulQA dataset and six multiple-choice datasets demonstrate that our method outperforms baselines. Our code will be released at `https://github.com/gjw185/FASB`.

## 1 Introduction

Large language models (LLMs) (Touvron et al., 2023; Brown et al., 2020) have achieved great success in text generation. However, the generated text still contains harmful information, hallucinations, and other misleading content. Therefore, controlling LLMs to produce trustworthy, reliable, and other desired outputs remains a challenge. Existing methods often use instruction tuning (Wei et al., 2022), Reinforcement Learning from Human Feedback (RLHF) (Bai et al., 2022), and prompt engineering (Brown et al., 2020) to control LLMs. Unfortunately, these methods often require large-scale datasets and expensive fine-tuning costs to achieve desired results.

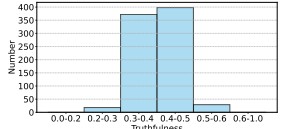

Figure 1: The truthfulness probability distribution of questions in the TruthfulQA dataset as detected by the classifier.

Recently, activation steering or representation engineering (Zou et al., 2023; Rimsky et al., 2024; Li et al., 2023; Turner et al., 2023; Cheng et al., 2025) has been proposed to control the outputs of LLMs by directly modifying their internal activations during inference, thereby avoiding the high cost associated

---

[*]Equal Contribution.
[†]Corresponding Author.

39th Conference on Neural Information Processing Systems (NeurIPS 2025).

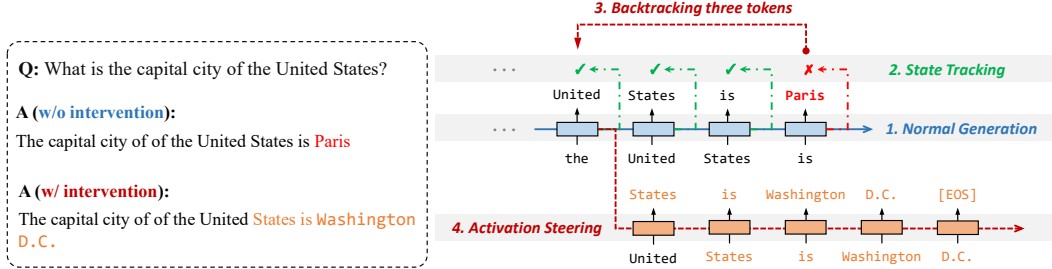

Figure 2: The overview of flexible activation steering with backtracking framework.

with large-scale data collection and fine-tuning. This technique constructs steering vectors from positive and negative samples to make targeted modifications to the activations of the LLM, enabling more precise control over its output.

Existing methods often apply interventions indiscriminately to all generations (Rimsky et al., 2024; Wang et al., 2024c; Li et al., 2023) or determine whether and how strongly to intervene based solely on the question (Lee et al., 2025), which limits the accurate assessment of the intervention strength. First, different generations should receive different levels of intervention. If a generation does not deviate, no intervention is necessary. If it deviates significantly, a stronger intervention should be applied. For example, when LLM is asked "Where is the capital of the United States?". If the response is "The capital of the United States is Washington, D.C.", no intervention is necessary, as the answer is correct. If the response is "Paris", it often requires a stronger intervention compared to "New York". Second, as each LLM generates different responses to the same question, question-only probing is challenging[1] and requires data collection for each LLM, resulting in high overhead. On the other hand, training a classifier on concatenated questions and answers often struggles to directly probe the question to determine whether and how strongly to intervene. As shown in Figure 1, the predicted truthfulness probabilities for questions in the TruthfulQA dataset are concentrated between 0.3 and 0.5, making fine-grained decisions difficult.

In this paper, we propose a **F**lexible **A**ctivation **S**teering with **B**acktracking (**FASB**) method, as shown in Figure 2. Unlike existing methods, **FASB** tracks the internal states of the LLM after each normal generation step, taking both the question and the generated content into account. In this way, **FASB** can dynamically determine whether intervention is necessary and the intervention strength based on the degree of deviation in the generation. Specifically, FASB employs two methods to identify internal states that are consistent with the desired behavior and derive the steering vector and classifier for state tracking. Considering that intervening after detecting a deviation from the desired behavior is often too late, we further propose the backtracking mechanism. The backtracking mechanism steps back a few tokens and performs activation steering to regenerate them, steering the generation toward the desired direction, with the intervention strength determined by the classifier.

Our main contributions are as follows:

- We propose a flexible activation steering with backtracking that dynamically determines both the necessity and strength of intervention by tracking the internal states of the LLMs.
- We propose a backtracking mechanism to step back a few tokens in order to apply intervention and regenerate them.
- We conduct extensive experiments on the TruthfulQA dataset and six multiple-choice datasets to demonstrate the effectiveness of our method.

## 2 Related Work

Activation steering or representation engineering (Zou et al., 2023; Rimsky et al., 2024; Li et al., 2023; Turner et al., 2023; Leong et al., 2023; Wang et al., 2025a,b) uses steering vectors to directly modify the activations of LLMs during inference to control their outputs in a desired direction. Activation steering preserves the general capabilities of LLMs while avoiding the expensive costs

---

[1]Questions in some domains may be relatively simple and general, such as those related to safety.

**(1) Heads Anchoring and Steering Vectors Inducing**

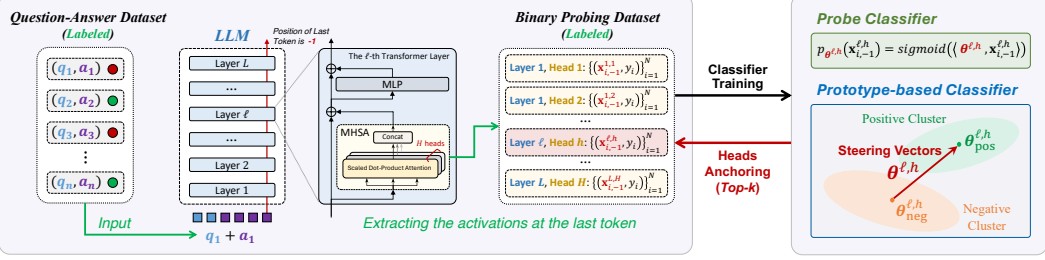

**(2) Generation with Flexible Steering and Backtracking**

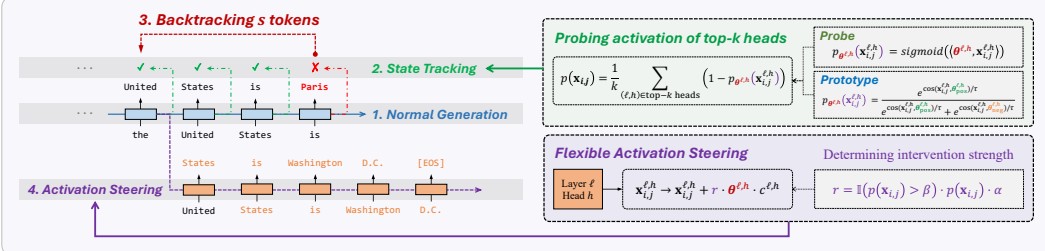

Figure 3: The framework of flexible activation steering with backtracking.

of high-quality data collection and fine-tuning (Zhang et al., 2025; Zhong et al., 2025; Shen et al., 2025).

The pioneering work (Zou et al., 2023; Turner et al., 2023; Wang et al., 2024a) creates and compares two prompts to obtain the steering vector. ITI (Li et al., 2023) uses the probe tool on a contrastive question-answering dataset to identify a set of attention heads associated with truthfulness. During inference, ITI modifies the activations of all subsequent generations in directions associated with truthfulness. Truth Forest (Chen et al., 2024) further employs multiple orthogonal probes to extract several truthful steering vectors, thereby improving performance. ACT (Wang et al., 2024b) uses clustering to construct multiple steering vectors and proposes an adaptive steering strength. CAA (Rimsky et al., 2024) computes steering vectors by averaging the difference between pairs of positive and negative examples. During inference, these steering vectors are added into the residual stream with a chosen steering strength at all token positions after the prompt to control the direction. ORTHO (Arditi et al., 2024) also averages the difference between pairs of positive and negative to compute the steering vector and performs directional ablation using the opposite direction of the steering vector to guide the model toward the desired behavior. SADI (Wang et al., 2024c) utilizes activation differences in contrastive pairs to precisely identify intervention position and dynamically steers model behavior by scaling element-wise activations. In addition, some works have explored the use of activation steering in instruction-following (Stolfo et al., 2024), in-context learning (Liu et al., 2024), and differentially private (Goel et al., 2025).

## 3 Method

Our method consists of two steps to flexibly steer model behavior, as illustrated in Figure 3. In the first step, we employ two alternative methods to identify the attention heads for intervention and to derive both the steering vector and the classifier. In the second step, the classifier is used for state tracking to determine whether intervention is necessary and to adaptively set intervention strength. We further propose a backtracking mechanism that allows the LLMs to regenerate tokens that deviate from the desired behavior. The full procedure can be found in Algorithm 1 in the Appendix.

### 3.1 Heads Anchoring and Steering Vectors Inducing

The first step is to use the classifier to identify the attention heads related to the desired behavior and to construct the steering vector. We use the **Probe** method to identify attention heads and induce the classifiers and steering vectors. In the Appendix A, we present an alternative **Prototype** method.

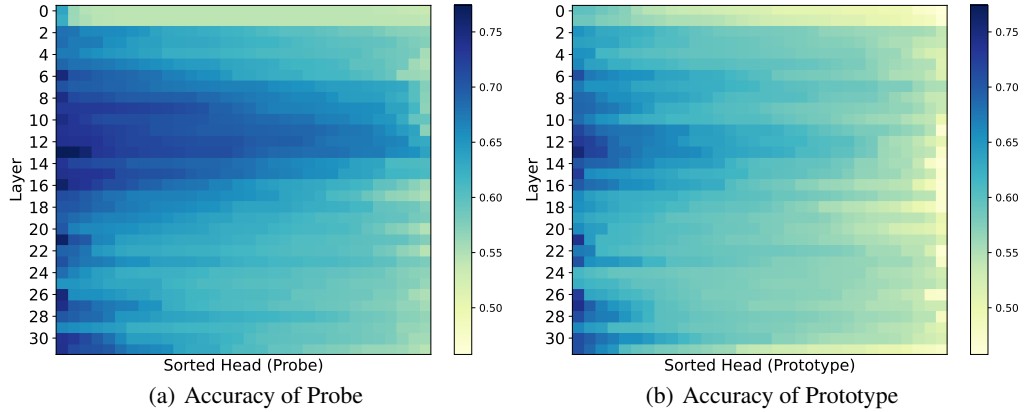

|   |   |
|---|---|
| (a) Accuracy of Probe | (b) Accuracy of Prototype |

Figure 4: Accuracies on the validation set of TruthfulQA dataset for all heads in all layers in LLaMA2-7B-CHAT, sorted row-wise by accuracy. Darker blue represents higher accuracy.

Since the **Probe** method uses a trainable classifier and typically achieves higher accuracy, we focus only on the **Probe** method in the section.

**Probe**    Probe method employs standard probing tools to identify attention heads. The idea behind probing tools is to train a lightweight classifier (probe) on the activations of attention heads to identify the relevant heads. Subsequently, we select the top-$k$ heads with the highest accuracy on the validation set for intervention, as they can effectively separate the samples and are more aligned with the desired behavior.

Specifically, we concatenate the question and answer from the labeled dataset and extract the activations at the last token to collect a binary probing dataset $\{(\mathbf{x}_{i,\text{-}1}^{\ell,h}, y_i)\}_{i=1}^N$ for each head in each layer, where $\mathbf{x}_{i,\text{-}1}^{\ell,h}$ denotes the activation of $h$-th attention head in the $\ell$-layer at the last token, and $y_i \in \{0,1\}$ denotes the label. The probe takes the following form:

$$p_{\boldsymbol{\theta}^{\ell,h}}(\mathbf{x}_{i,\text{-}1}^{\ell,h}) = \text{sigmoid}(\langle \boldsymbol{\theta}^{\ell,h}, \mathbf{x}_{i,\text{-}1}^{\ell,h} \rangle) \tag{1}$$

where $\langle,\rangle$ denotes dot product, and $\boldsymbol{\theta}^{\ell,h}$ denotes the probe parameter of $h$-th attention head in the $\ell$-layer. Since the probes can effectively separate positive and negative examples, we use the parameters of probe $\boldsymbol{\theta}^{\ell,h}$ as the steering vector.

As shown in Figure 4, a subset of heads is strongly related to the truthfulness, and the optimal heads are relatively evenly distributed across all layers. Therefore, performing fine-grained intervention only on these heads is more targeted and helps minimize disruption to the model's overall behavior. It is worth noting that our method can also intervene in the outputs of the MLP module or the outputs of layers.

### 3.2    Generation with Flexible Steering and Backtracking

In the second step, after generating each token, the classifier is used for state tracking to determine whether the generation deviates from the desired behavior, allowing for flexible intervention. If a deviation is detected during the generation process, it often indicates that the previously generated tokens have deviated from the desired behavior. We backtrack a few tokens to regenerate them and apply steering vectors to adaptively intervene in all subsequent tokens. The backtracking mechanism enables the correction of deviated tokens and helps steer the model toward the desired behavior. If no deviation is detected, the model continues generating normally.

**State Tracking**    After generating the $j$-th token, the classifier probes the $j$-th token's activation of the LLM to assess whether the current generation deviates from the desired behavior and determines the intervention strength. Notably, state tracking and probe share the same goal: determining whether the current response deviates from the desired behavior. Specifically, we average the pre-

diction probabilities from the top-$k$ selected heads to assess the deviation:

$$p(\mathbf{x}_{i,j}) = \frac{1}{k} \sum_{(\ell,h) \in \text{top-}k \text{ heads}} (1 - p_{\boldsymbol{\theta}^{\ell,h}}(\mathbf{x}_{i,j}^{\ell,h})) \tag{2}$$

where $p(\mathbf{x}_{i,j})$ denotes the deviation probability after generating $j$-th token for $\mathbf{x}_i$, and $\mathbf{x}_{i,j}^{\ell,h}$ represents the activation of $j$-th generated token for $\mathbf{x}_i$ at the $\ell$-th layer and the $h$-th head. The process is highly lightweight, as the hidden states are already generated during the generation process.

**Backtracking**  When the deviation probability exceeds the threshold, we consider the generation to deviate from the desired behavior, and intervention should be applied. An important issue is that intervening only after a deviation is detected is often too late, as it can only affect the content generated subsequently. To address this issue, we propose a backtracking mechanism to regenerate the previous deviated tokens. If a deviation is detected after generating the $j$-th token, the backtracking mechanism retains only the first $j$-$s$ tokens and regenerates the subsequent tokens, where $s$ is a hyperparameter that controls the number of tokens to backtrack. Compared to generating from scratch, the backtracking mechanism only requires generating an additional $s$ tokens, making the overhead lightweight. Subsequently, we apply intervention and regenerate the content after the ($j$-$s$)-th token to prevent further deviation.

**Activation Steering**  We further introduce an adaptive intervention strength for activation steering, where the strength is determined by the degree of deviation calculated by the classifier and is proportional to it. The intervention strength is defined as follows:

$$r = \mathbb{I}(p(\mathbf{x}_{i,j}) > \beta) \cdot p(\mathbf{x}_{i,j}) \cdot \alpha \tag{3}$$

where $\alpha$ and $\beta$ are hyperparameters that control the intervention strength and the threshold for deviation probability, respectively. An appropriately sized $\alpha$ can steer the LLM toward the desired behavior without compromising generation quality. When $\alpha$ is too large, it degrades generation quality, whereas when it is too small, it fails to provide effective steering. An appropriately sized $\beta$ allows intervention when deviations occur. An excessively large $\beta$ may miss necessary interventions, while an excessively small one may cause over-intervention.

Since we intervene in the output of selected heads, the MHSA with intervention can be formulated as follows:

$$\mathbf{h}_{i,j-s+1}^{\ell} = \text{concat}(\mathbf{x}_{i,j-s+1}^{\ell,1} + r\boldsymbol{\theta}^{\ell,1}c^{\ell,1}, \cdots, \mathbf{x}_{i,j-s+1}^{\ell,H} + r\boldsymbol{\theta}^{\ell,H}c^{\ell,H})\mathbf{W}^{\ell,O} \tag{4}$$

where $\mathbf{h}_{i,j-s+1}^{\ell}$ represents the output of $\ell$-th MHSA at the ($j$-$s$+1)-th token, $\mathbf{x}_{i,j-s+1}^{\ell,H}$ represents the output of self-attention for $H$-th head in $\ell$-th layer, and $\mathbf{W}^{\ell,O}$ is output projection matrix in $\ell$-th layer. $c^{\ell,H}$ is a binary scalar that equals 1 for the selected top-$k$ heads and 0 for the unselected ones.

# 4  Experiments

## 4.1  Experimental Settings

**Datasets and Evaluation Metrics**  We conduct experiments on open-ended generation tasks and multiple-choice tasks. TruthfulQA (Lin et al., 2022) dataset includes open-ended generation task and multiple-choice task. For the open-ended generation, we employ two fine-tuned LLMs to judge whether the answer is truthful[2] and informative[3], denoted as True (%) and Info (%) respectively, while the product True*Info (%) serves as the primary metric. For the multiple-choice tasks, we use datasets: COPA (Gordon et al., 2012), StoryCloze (Mostafazadeh et al., 2016), NLI (Bowman et al., 2015), MMLU (Hendrycks et al., 2021), SST2 (Socher et al., 2013), and Winogrande (Sakaguchi et al., 2020), with response formats ranging from 2-way to 4-way choices. We use multiple-choice accuracy (MC) to evaluate.

**Implementation Details**  In the Probe method, for the TruthfulQA dataset, we intervene using the top-24 heads, set the threshold range to [0.4, 0.5], the number of backtracking steps to 10, and search

---

[2]https://huggingface.co/allenai/truthfulqa-truth-judge-llama2-7B
[3]https://huggingface.co/allenai/truthfulqa-info-judge-llama2-7B

Table 1: Results on TruthfulQA open-ended generation (True*Info %) and multiple-choice tasks (MC %).

| Methods | Open-ended Generation | | | Multiple-Choice | | |
|---|---|---|---|---|---|---|
| | True (%) | Info (%) | True*Info (%) | MC1 (%) | MC2 (%) | MC3 (%) |
| **Baseline** | 66.83 | 99.51 | 66.50 | 33.41 | 51.07 | 24.76 |
| **ITI** | 94.49 | 80.55 | 76.11 | 38.31 | 57.15 | 30.53 |
| **CAA** | 71.60 | 83.84 | 60.03 | 34.03 | 52.76 | 25.62 |
| **ORTHO** | 67.94 | 90.09 | 61.21 | 36.23 | 52.88 | 26.12 |
| **CAST** | 67.69 | 86.17 | 58.33 | 33.90 | 51.17 | 25.01 |
| **ACT** | - | - | - | 28.80 | 45.20 | - |
| **SADI-HEAD** | 77.72 | 98.53 | 76.58 | 35.90 | 54.65 | 26.99 |
| **Probe (Ours)** | 93.88 | 85.81 | **80.56** | **48.71** | **66.58** | **41.95** |

Table 2: Results (MC %) on six multiple-choice tasks.

| Methods | COPA | StoryCloze | NLI | MMLU | SST2 | Winogrande | AVG |
|---|---|---|---|---|---|---|---|
| **Baseline** | 64.4 | 60.2 | 63.5 | 60.2 | 92.2 | 50.2 | 65.1 |
| **ITI** | 66.6 | 59.7 | 64.3 | 60.2 | 92.3 | 51.5 | 65.8 |
| **CAA** | 66.6 | 63.5 | 64.9 | 62.6 | 92.2 | 50.9 | 66.8 |
| **ORTHO** | 65.2 | 60.2 | 63.1 | **63.8** | 92.4 | 50.4 | 65.8 |
| **SADI** | 65.4 | 60.5 | 65.1 | 61.8 | 92.3 | 51.8 | 66.1 |
| **Probe (Ours)** | **90.0** | **93.5** | **80.0** | 62.4 | **92.8** | **54.1** | **78.8** |

for the intervention strength in the range of [40, 80] with a step size of 10. For the six multiple-choice tasks, our threshold search range is [0.3, 0.4, 0.5, 0.6], the intervention strength search range is [0, 250] with a step size of 10, and the number of backtracking steps is 10.

## 4.2 Baselines

We compare our model with the following baselines to show its effectiveness. **Baseline** directly uses the original LLaMA2-7B-CHAT model to generate text. **ITI** (Li et al., 2023) identifies a set of attention heads with high linear probing accuracy and shifts activations of all subsequent generations following the user's prompt along these probe-correlated directions. **CAA** (Rimsky et al., 2024) computes the steering vector by averaging the difference between pairs of positive and negative examples. During inference, these steering vectors are added at all generations with a coefficient. **ORTHO** (Arditi et al., 2024) uses the same reversed steering vector for directional ablation, steering the generation toward the desired direction. **CAST** (Lee et al., 2024) computes the condition vector and behavior vectors using PCA. During the inference phase, it uses the condition vector to make judgments, enabling dynamic intervention in generations. **ACT** (Wang et al., 2024b) constructs multiple classifiers through clustering and uses these classifiers to dynamically intervene in responses from different directions. **SADI** (Wang et al., 2024c) utilizes activation differences in contrastive pairs to precisely identify intervention position and dynamically steer model behavior by scaling element-wise activations.

## 4.3 Results

**Results on TruthfulQA** In Table 1, the Probe method demonstrates superior performance compared to the baselines on the TruthfulQA dataset. Compared to the ITI method, our Probe method does not require additional training on top of ITI and has achieved performance far exceeding that of the ITI method. This is because ITI applies the same intervention strength indiscriminately to all generations. This makes it impossible to assign adaptive intervention strength and may cause originally correct answers to be incorrectly altered due to the intervention. In contrast, our method adaptively determines the intervention strength and does not intervene in correct generations. Compared with other dynamic intervention methods such as ORTHO, CAST, ACT, and SADI, it can be noted that SADI also achieves good performance, which demonstrates that the dynamic steering vector is effective. ORTHO and CAST are mainly designed for safety-related scenarios, where it is relatively

Table 3: Ablation study on TruthfulQA dataset.

| Methods | True*Info | MC1 | MC2 |
|---|---|---|---|
| **Probe** | 80.56 | **48.71** | **66.58** |
| *Intervention Strength:* | | | |
| **All fixed strength** | 76.11 | 38.31 | 57.15 |
| **w/o Adaptive** | **82.08** | 42.96 | 62.06 |
| *Intervention Position:* | | | |
| **w/o Backtracking** | 62.11 | 35.01 | 53.55 |
| **After Question** | 72.55 | 41.86 | 59.88 |

easy to determine whether a query is harmful based solely on the query information. However, in domains such as truthfulness and faithfulness, it is difficult to anticipate whether the generated content will deviate based solely on the query, resulting in poor performance. Compared with these dynamic intervention methods, the superiority of our method can be further demonstrated.

**Results on Multiple-choice** In Table 2, our method achieves consistently promising results across all datasets, and Probe method achieves the best performance. This demonstrates that adaptive intervention and using probes to select heads remain effective on multiple-choice datasets. Notably, CAA achieves better results than ITI, suggesting that layer-wise intervention can sometimes lead to better performance. SADI also performs well in multiple-choice tasks, which demonstrates the generalizability of the dynamic steering vector.

## 4.4 Ablation Study

We conduct an ablation study to show the effectiveness of the proposed components in Table 3.

We first explore the ablation of intervention strength. "All fixed strength" refers to the results of applying the same intervention strength to all samples. "w/o Adaptive" refers to the results of applying the same intervention strength to samples that meet the intervention criteria. By comparing the results of "w/o Adaptive" with "All fixed strength", it is demonstrated that intervening only on samples that deviate from the desired behavior can improve performance. Comparing the results of the Probe with "w/o Adaptive" demonstrates the effectiveness of dynamic intervention strength.

We then explore the ablation of the intervention position. "w/o Backtracking" means no backtracking is performed. "After Question" indicates that the representation of the question was used to decide whether to intervene and intervention strength. Comparing the results of the Probe with "w/o Backtracking" shows the necessity of the backtracking operation. The True*Info metric of "w/o Backtracking" is even lower than the baseline. This indicates that intervening after detecting the deviation from the desired behavior is too late. Comparing the results of the Probe with "After Question" indicates that relying solely on the hidden states of the question part to judge whether the subsequent response deviates from the desired behavior is insufficient. It is necessary to make judgments after generating part of the response.

## 4.5 Generalizability across More Truthful Benchmarks

We further investigate whether our method can generalize beyond the TruthfulQA benchmark. Specifically, we directly evaluate the classifier and steering vectors obtained from TruthfulQA on two datasets related to real-world truth, including Natural Questions (Kwiatkowski et al., 2019) and TriviaQA (Joshi et al., 2017). The Natural Questions dataset consists of 3,610 real queries issued to the Google search engine, annotated with answers and supporting Wikipedia pages. TriviaQA includes 95K question-answer pairs annotated by trivia enthusiasts. Following Li et al. (2023), all benchmarks are presented in a multiple-choice format.

The results indicate that the Probe method outperforms the baseline and the ITI on both benchmarks, as shown in Table 4. This suggests that employing the Probe method does not undermine the model's performance in out-of-distribution truthful domains; Instead, it enhances the model's performance, particularly in domains closely related to the real-world truth. This indicates that the classifier and

Table 4: MC2 on the Natural Questions and TriviaQA multiple-choice datasets using LLaMA2-7B-CHAT as the baseline.

| Methods | Natural Questions | TriviaQA |
|---------|-------------------|----------|
| **Baseline** | 49.54 | 61.22 |
| **ITI** | 56.50 | 66.49 |
| **Probe** | **59.25** | **67.55** |

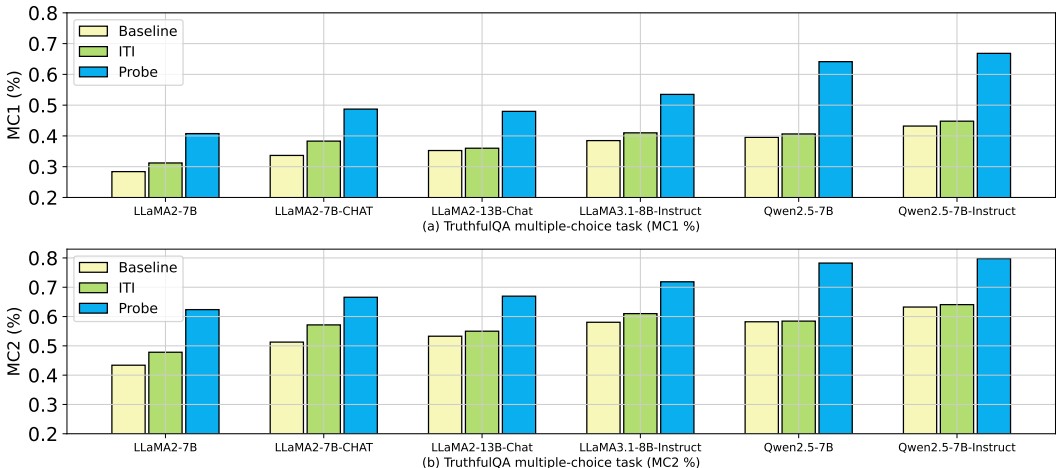

Figure 5: The performance of various LLMs on the TruthfulQA benchmark.

steering vector learned on the TruthfulQA dataset are not domain-specific, but rather general real-world truth features.

### 4.6 Generality across More LLMs

To evaluate the generality across various LLMs, we apply the Probe method to 6 LLMs, including LLaMA2 Touvron et al. (2023), LLaMA3.1 (Grattafiori et al., 2024), and Qwen2.5 (Yang et al., 2024). The performance of baseline, ITI, and Probe on the TruthfulQA benchmark in Figure 5.

The probing method effectively improves performance across 6 LLMs, demonstrating that our approach can generalize to models of different sizes, architectures, and whether or not they have been instruction-tuned. Notably, our method yields significant improvements of 24.61 and 20.03 in MC1 and MC2, respectively, on Qwen2.5-7B. Compared with the ITI method, our method achieves better performance on all 6 LLMs. On Qwen2.5-7B, there is an improvement of 23.50 on the MC1 and an improvement of 19.81 on the MC2. It is demonstrated that our method outperforms the ITI method in terms of performance across different LLMs. Scaling up the model size often results in better performance, indicating that it has acquired more knowledge related to truthfulness. Instruction-tuned models achieve better performance than non-instruction-tuned ones, indicating that instruction tuning helps enhance the truthfulness of the model.

### 4.7 Model Analysis

**Effects of intervention strength and head number** In Figure 6, we present the results of our method on the TruthfulQA dataset. We sweep two hyperparameters to control the intervention: the number of identified attention heads and the intervention strength.

The truthful metrics MC1, MC2, and True show a trend where the results continuously improve as both the strength and the number of attention heads increase. This is because, with the continuous increase in the number of heads and the strength of intervention, the internal truthfulness of the model is constantly increasing, making the model more inclined to choose and respond with more truthful answers. It is worth noting that our method can be applied with greater strength, as it does not intervene with all samples, and the intervention strength is adaptive.

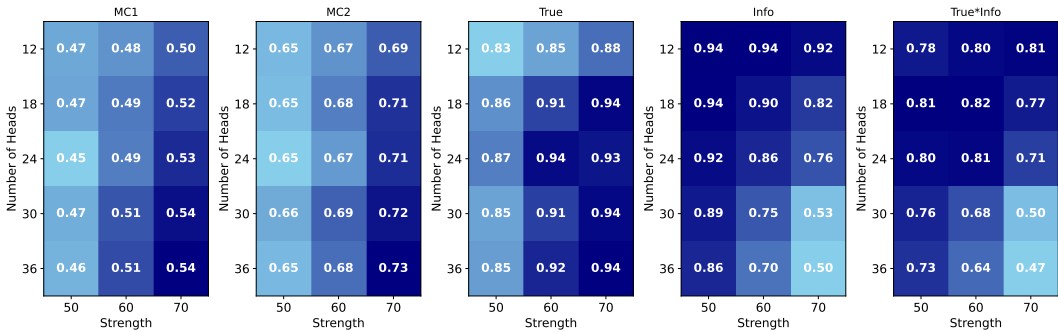

Figure 6: Results with varying intervention strength and numbers of attention heads on the Truth-fulQA dataset with LLaMA2-7B-CHAT.

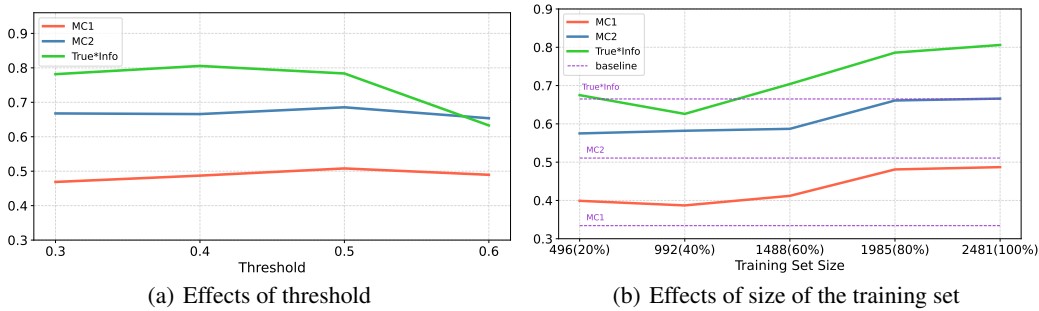

| (a) Effects of threshold | (b) Effects of size of the training set |
|---|---|

Figure 7: Effects of the threshold and the size of the training set on TruthfulQA dataset.

The Info metric decreases as the number of heads increases and the intervention strength increases. This is because, with the continuous increase in intervention strength, the model internally encodes more truthfulness, leading the model to be more inclined to respond with answers that are truthful but lack informativeness. Due to the mutual restraint between the True metric and the Info metric, the True*Info metric achieves highest values when the True metric and the Info metric are relatively balanced, i.e., the number of heads is 18 and the strength is 60. Therefore, the optimal hyperparameters for MC2 and True*Info are not consistent.

**Effects of threshold** To investigate the impact of different thresholds $\beta$, we conduct experiments with thresholds set at 0.3, 0.4, 0.5, and 0.6. When $\beta$ is 0.3, 0.4, or 0.5, the overall performance does not vary significantly. Among them, True*Info achieves the best performance at 0.4, while MC1 and MC2 perform best at 0.5. When the threshold is set to 0.6, True*Info drops sharply, which may be due to excessive intervention.

**Effects of training set size** To better investigate the impact of the size of the used dataset on the performance, we conduct experiments by utilizing 20%, 40%, 60%, 80%, and 100% of the original dataset. As the size of the dataset increases, the metrics MC1, MC2, and True*Info generally exhibit an upward trend, essentially reaching their maximum values after utilizing approximately 80% of the data, as shown in Figure 7 (b). Moreover, when only 20% of the data is used, *i.e.*, 496 samples, all three metrics can exceed the baseline, with more significant improvements observed in the MC1 and MC2 metrics. This indicates that our method does not require a large dataset. As the amount of data increases, performance improvements become both more consistent and more significant.

## 5 Conclusion

In this paper, we propose a Flexible Activation Steering with Backtracking framework. It dynamically decides whether and how strongly to intervene by probing the internal states of the LLM. Specifically, FASB first identifies attention heads that are consistent with the desired behavior and

derive the steering vector and classifier. Then, FASB uses classifiers to dynamically determine whether and how strongly to intervene by probing the internal states of the LLM. Finally, we further propose a backtracking mechanism to regenerate in order to avoid deviating from the deviated behavior. Experimental results on the TruthfulQA dataset and six multiple-choice datasets to demonstrate the effectiveness of our method.

## Acknowledgments and Disclosure of Funding

We would like to thank the anonymous reviewers for their insightful comments. This work is supported by the JiangSu Natural Science Foundation under Grant No. BK20251989; the National Natural Science Foundation of China under Grants Nos. 62441225, 61972192, 62172208; the Fundamental Research Funds for the Central Universities under Grant No. 14380001. This work is partially supported by Collaborative Innovation Center of Novel Software Technology and Industrialization.

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

Table 5: Results on TruthfulQA open-ended generation (True*Info %) and multiple-choice tasks (MC %).

| Methods | Open-ended Generation | | | Multiple-Choice | | |
|---|---|---|---|---|---|---|
| | True (%) | Info (%) | True*Info (%) | MC1 (%) | MC2 (%) | MC3 (%) |
| **Baseline** | 66.83 | 99.51 | 66.50 | 33.41 | 51.07 | 24.76 |
| **ITI** | 94.49 | 80.55 | 76.11 | 38.31 | 57.15 | 30.53 |
| **CAA** | 71.60 | 83.84 | 60.03 | 34.03 | 52.76 | 25.62 |
| **ORTHO** | 67.94 | 90.09 | 61.21 | 36.23 | 52.88 | 26.12 |
| **CAST** | 67.69 | 86.17 | 58.33 | 33.90 | 51.17 | 25.01 |
| **ACT** | - | - | - | 28.80 | 45.20 | - |
| **SADI-HEAD** | 77.72 | 98.53 | 76.58 | 35.90 | 54.65 | 26.99 |
| **Probe (Ours)** | 93.88 | 85.81 | 80.56 | **48.71** | **66.58** | **41.95** |
| **Prototype (Ours)** | 88.37 | 94.98 | **83.94** | 46.14 | 64.30 | 37.07 |

Table 6: Results (MC %) on six multiple-choice tasks.

| Methods | COPA | StoryCloze | NLI | MMLU | SST2 | Winogrande | AVG |
|---|---|---|---|---|---|---|---|
| **Baseline** | 64.4 | 60.2 | 63.5 | 60.2 | 92.2 | 50.2 | 65.1 |
| **ITI** | 66.6 | 59.7 | 64.3 | 60.2 | 92.3 | 51.5 | 65.8 |
| **CAA** | 66.6 | 63.5 | 64.9 | 62.6 | 92.2 | 50.9 | 66.8 |
| **ORTHO** | 65.2 | 60.2 | 63.1 | **63.8** | 92.4 | 50.4 | 65.8 |
| **SADI** | 65.4 | 60.5 | 65.1 | 61.8 | 92.3 | 51.8 | 66.1 |
| **Probe (Ours)** | **90.0** | **93.5** | **80.0** | 62.4 | 92.8 | 54.1 | **78.8** |
| **Prototype (Ours)** | 87.8 | 86.1 | 73.7 | 62.2 | **93.1** | **54.7** | 76.3 |

# A  Prototype Method

**Prototype**  Prototype method directly constructs two prototypes to achieve the above goal. Specifically, we compute the average activation within each class to obtain the prototype representations of $h$-th attention head in the $\ell$-th layer, as follows:

$$\boldsymbol{\theta}_{\text{pos}}^{\ell,h} = \frac{1}{N_{pos}} \sum_{i=1}^{N_{pos}} \mathbb{I}(y_i = 1)\mathbf{x}_{i,\text{-}1}^{\ell,h}, \quad \boldsymbol{\theta}_{\text{neg}}^{\ell,h} = \frac{1}{N_{neg}} \sum_{i=1}^{N_{neg}} \mathbb{I}(y_i = 0)\mathbf{x}_{i,\text{-}1}^{\ell,h} \tag{5}$$

where $\boldsymbol{\theta}_{\text{pos}}^{\ell,h}$ and $\boldsymbol{\theta}_{\text{neg}}^{\ell,h}$ represent the prototypes of the positive and negative classes, respectively. $\mathbb{I}$ denotes indicator function, and $N_{pos}$ and $N_{neg}$ represent the numbers of positive and negative samples, respectively. Then, we can use the softmax function over cosine similarities between activation and prototypes to define a classifier:

$$p_{\boldsymbol{\theta}^{\ell,h}}(\mathbf{x}_{i,\text{-}1}^{\ell,h}) = \frac{\exp(\cos(\mathbf{x}_{i,\text{-}1}^{\ell,h}, \boldsymbol{\theta}_{\text{pos}}^{\ell,h})/\tau)}{\exp(\cos(\mathbf{x}_{i,\text{-}1}^{\ell,h}, \boldsymbol{\theta}_{\text{pos}}^{\ell,h})/\tau) + \exp(\cos(\mathbf{x}_{i,\text{-}1}^{\ell,h}, \boldsymbol{\theta}_{\text{neg}}^{\ell,h})/\tau)} \tag{6}$$

where $\tau$ denotes the temperature. We also use the classifier to anchor the top-$k$ heads with high accuracy for intervention. Finally, we directly use the mean difference between positive and negative prototypes as the steering vector:

$$\boldsymbol{\theta}^{\ell,h} = \boldsymbol{\theta}_{\text{pos}}^{\ell,h} - \boldsymbol{\theta}_{\text{neg}}^{\ell,h} \tag{7}$$

It is worth noting that the prototype method is training-free. It only involves averaging the activations of the training set to obtain prototype vectors for constructing the classifier.

**Implementation Details**  In the Prototype method, for the TruthfulQA dataset, we intervene using the top-24 heads, set the threshold to 0.5, the number of backtracking steps to 10, the temperature to 0.1, and search for the intervention strength in the range of [25, 55] with a step size of 10. For the six multiple-choice tasks, the threshold is 0.5, the number of backtracking steps is 10, the temperature is 0.1, and the intervention strength search range is [0, 400] with a step size of 10.

**Results**  In Table 5, the Prototype method also demonstrates superior performance compared to the baselines on the TruthfulQA dataset. Compared with the ITI and ACT methods that require training a linear classifier, the Prototype method achieves improvements of 7.83% and 17.34% on MC1, and 7.15% and 19.1% on MC2,

respectively. This demonstrates the generality of our method. It is worth noting that the Probe method performs better on the True and multiple-choice metrics, while the Prototype method achieves higher performance on the Info and True*Info metrics. This may be because the trained probe is more closely related to truthfulness.

In Table 6, the Prototype method also achieves better performance than baselines on six multiple-choice tasks. On the COPA dataset and the StoryCloze dataset, the Prototype method achieves improvements of 23.4% and 25.9% compared with the model without intervention, while other methods only achieve marginal improvements. This further demonstrates its effectiveness.

## B  Algorithm

The algorithm consists of two main processes. The first step is **Heads Anchoring and Steering Vectors Inducing**, which determines the heads that need to be intervened, the classifiers, and the steering vectors. The second step is **Generation with Flexible Steering and Backtracking**, which involves using classifiers to evaluate the internal activations of LLMs. When the internal activations exceed a certain threshold, a backtracking operation will be performed. After the backtracking, the activations will be regenerated by adding the activation vectors. Since our method also backtracks to the beginning when the number of generated tokens is less than $s$, we start tracking from the $s$-th token, as shown in Line 14 of the algorithm.

---

**Algorithm 1** The overall flow of FASB

---

**Require:** LLM, dataset $\mathcal{D}$, intervention strength $\alpha$, threshold $\beta$, backtracking number $s$, maximum number of generated tokens $M$

1: **Step1: Heads Anchoring and Steering Vectors Inducing**
2: **if** Probe **then**
3:      Train classifier $p_{\boldsymbol{\theta}^{\ell,h}}$ on dataset $\mathcal{D}$;          ▷ Equation 1
4:      Use the parameters of the probe as the steering vector $\boldsymbol{\theta}^{\ell,h}$;
5: **else if** Prototype **then**
6:      Construct two prototypes from the dataset $\mathcal{D}$;          ▷ Equation 5
7:      Obtain the classifier $p_{\boldsymbol{\theta}^{\ell,h}}$ ;          ▷ Equation 6
8:      Obtain the steering vector $\boldsymbol{\theta}^{\ell,h}$;          ▷ Equation 7
9: **end if**
10: **Step2: Generation with Flexible Steering and Backtracking**
11: **for** $j = 1$ to $M$ **do**
12:      Generate the $j$-th token;
13:      Evaluate the current activation using the classifier;          ▷ Equation 2
14:      **if** $p(\mathbf{x}_{i,j}) > \beta$ and $j \geq s$ **then**
15:          Backtrack $s$ tokens;
16:          Calculate the intervention strength;          ▷ Equation 3
17:          **for** $k = (j\text{-}s+1)$ to $M$ **do**
18:              Intervene on the current activation;          ▷ Equation 4
19:              Generate the $k$-th token;
20:          **end for**
21:          break;
22:      **end if**
23: **end for**

---

## C  Effects of the Number of Tokens for Backtracking

In order to better investigate the impact of the backtracking number on our method, we conduct experiments on the TruthfulQA dataset.

As shown in Figure 8, we investigate the effect of different token numbers for backtracking (*i.e.*, 2, 5, 10, 20) on the MC1, MC2, and True*Info metrics. MC1, MC2, and True*Info generally show an increasing trend with the increase in the number of backtracking steps. This is because, as the number of backtracking increases, our method can intervene in the model's internal activations earlier, thereby achieving improved performance.

## D  Fine-Grained Model Comparison

We further propose two models based on Probe for fine-grained analysis. The difference between them and Probe lies in the detection location and the point at which intervention begins, resulting in different overheads.

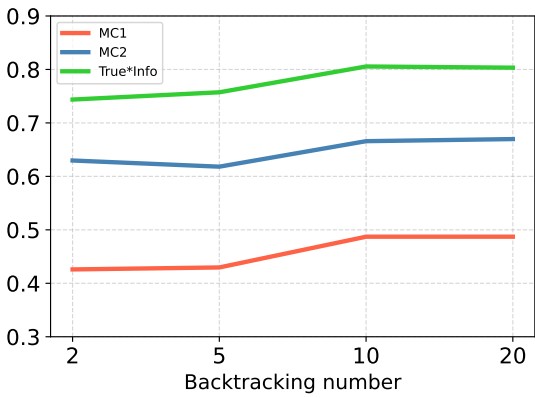

Figure 8: Effects of the number of tokens for backtracking.

Table 7: Comparison with two fine-grained models.

| Methods | True*Info | MC1 | MC2 |
|---------|-----------|-----|-----|
| **Probe** | 80.56 | 48.71 | 66.58 |
| **BTB** | 81.60 | 48.83 | **67.62** |
| **GCBB** | **81.96** | **50.67** | 67.48 |

(1) The first method directly **B**ack**T**racks to the **B**eginning (**BTB**). Specifically, when a deviation is detected during generation, BTB backtracks not just $s$ tokens for regeneration, but to the beginning. Our method generates $s$ additional tokens for texts that require backtracking, whereas this variant generates even more. (2) The second method performs detection after the **G**eneration is **C**omplete and then **B**acktracks to the **B**eginning (GCBB). Notably, this method incurs approximately double the overhead for texts that require backtracking.

As shown in the results in Table 7, GCBB usually achieves the best performance, and both BTB and GCBB outperform our proposed Probe method on the TruthfulQA dataset. This is because GCBB has access to the full output of the LLM, allowing it to better determine the intervention strength. However, GCBB inevitably introduces additional overhead. BTB may also sometimes require full regeneration. The advantage of Probe is that it achieves good performance while maintaining stable and relatively low additional overhead.

# E  Results across TruthfulQA Categories

TruthfulQA is split into 38 subcategories, including politics, language, education, psychology and others. We compare our method with the baseline method without intervention using the True*Info metric across all subcategories with 10 or more questions, where the subcategories are ranked in descending order based on their quantity within the dataset, as shown in Figure 9.

Our method demonstrates significant enhancement across most subcategories, with the overall performance improvement showing uniform distribution rather than concentration in particular domains, thereby validating its efficacy.

# F  Results on Multi-hop QA dataset

We conduct experiments on the more challenging multi-hop question answering WikiHop dataset (Welbl et al., 2018) to verify the effectiveness, as shown in Table 8. Our method achieves improvements of 5% and 4.3% on metrics MC1 and MC2, respectively. This further demonstrates the generalizability of our method.

Table 8: Results on the WikiHop dataset.

| Method | MC1(%) | MC2(%) |
|--------|--------|--------|
| LLaMA2-7B-CHAT | 45.20 | 44.03 |
| LLaMA2-7B-CHAT + Probe | **50.20 (+ 5.00)** | **48.33 (+4.30)** |

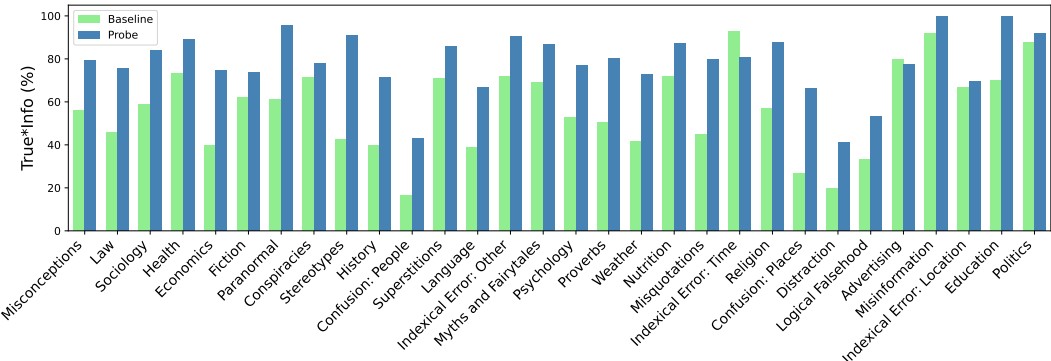

Figure 9: True*Info scores split across subcategories on LLaMA2-7B-CHAT, sorted by the difference between baseline and probe method. Subcategories with less than 10 questions are not shown.

# G   Generalizability on larger LLM

We further explore the effectiveness of our method on larger LLMs by using QWEN2.5-32B-Instruct on the TruthfulQA dataset in Table 9. Our method still achieves significant performance improvements, demonstrating its effectiveness on larger LLMs as well.

Table 9: Results on the Qwen2.5-32B-Instruct model.

| Method | MC1(%) | MC2(%) | MC3(%) |
|---|---|---|---|
| Qwen2.5-32B-Instruct | 50.00 | 66.35 | 38.75 |
| Qwen2.5-32B-Instruct +Probe | 69.00 | 76.67 | 59.59 |

# H   Deviation Positions Under Different Thresholds

The Table 10 presents the detected deviation positions under different thresholds on the TruthfulQA dataset. We observe that the threshold has a significant impact on the intervention position. When the threshold is high, the intervention positions tend to occur significantly later, and the interventions are less frequent. This is because the model needs to track more generated content to make a more accurate judgment about whether a deviation occurs.

Table 10: Distribution of tokens under different thresholds.

| Threshold $\beta$ | Position | | |
|---|---|---|---|
| | 0-10 | 10-20 | 20-50 |
| 0.4 | 304 | 94 | 9 |
| 0.5 | 166 | 159 | 75 |
| 0.6 | 21 | 109 | 222 |

# I   Results on Toxic Context Generation

Since our method can flexibly manipulate various behaviors, we further conduct experiments on the RealToxicityPrompts dataset (Gehman et al., 2020) for toxic content generation, as shown in Table 11. Our method can effectively steer LLMs to generate toxic content, demonstrating its generality. This also highlights the vulnerability of current alignment methods and the potential risks of our approach.

# J   Comprehensive Hyperparameter Analysis of strength $\alpha$ and threshold $\beta$

We further conduct a more comprehensive exploration of the impact of strength $\alpha$ and threshold $\beta$ on the TruthfulQA dataset.

Table 11: Results on the RealToxicityPrompts dataset.

| Method | ASR ($\uparrow$) |
|---|---|
| LLaMA2-7B-CHAT | 42.0 |
| LLaMA2-7B-CHAT + Probe | **46.4 (+4.4)** |

Table 12: Impact of strength.

| Strength $\alpha$ | MC1 (%) | MC2 (%) | True (%) | Info (%) | True*Info (%) |
|---|---|---|---|---|---|
| 0 (baseline) | 33.41 | 51.07 | 66.83 | 99.51 | 66.50 |
| 2 | 36.43 | 52.96 | 67.24 | 84.35 | 56.72 |
| 5 | 38.63 | 54.94 | 68.46 | 84.35 | 57.75 |
| 15 | 43.77 | 60.14 | 73.71 | 88.86 | 65.50 |
| 25 | 43.33 | 62.02 | 80.18 | 91.06 | 73.01 |
| 35 | 43.33 | 62.66 | 82.01 | 93.88 | 76.99 |
| 45 | 45.04 | 63.84 | 83.61 | 93.88 | 78.49 |
| 50 | 45.04 | 64.52 | 87.15 | 91.80 | 80.05 |
| 60 | 48.71 | 66.58 | 93.88 | 85.81 | 80.56 |
| 65 | 50.18 | 68.59 | 95.11 | 77.24 | 73.46 |
| 70 | 52.51 | 70.89 | 93.26 | 76.38 | 71.24 |
| 100 | 55.75 | 74.35 | 97.83 | 18.51 | 18.11 |
| 500 | 43.03 | 75.48 | 89.99 | 6.36 | 5.68 |

As shown in Table 12, when $\alpha$ is small (e.g., $\alpha = 2, 5$), it often fails to effectively improve the True metric and even leads to a decrease in the Info metric. This indicates that a low intervention strength fails to enhance truthfulness and even reduces informativeness. Conversely, an excessively large intervention strength (e.g., $\alpha = 100, 500$) reduces the informativeness of the generated text. For instance, when $\alpha = 500$, we observe that the Info metric drops to 6.36%, indicating that the generated text lacks informativeness. An appropriately sized $\alpha$ often enhances the truthfulness of the generated content while causing less information loss.

Then, we explore the impact of the threshold $\beta$. As shown in Table 13, when $\beta$ is too large (e.g., $\beta = 0.7, 0.8, 0.9$), many samples would not be intervened, leading to lower performance on the MC1, MC2, and True metrics. For instance, the performance on these three metrics with $\beta = 0.7, 0.8, 0.9$ is significantly lower than that with $\beta = 0.6$. When $\beta$ is too small (e.g., $\beta = 0, 0.2$), many samples would be backtracked and intervened at the first state judgment. In such cases, although performance may remain satisfactory due to our adaptive intervention strength, the method loses the flexibility to decide whether intervention is needed.

Table 13: Impact of threshold.

| Threshold $\beta$ | MC1 (%) | MC2 (%) | True (%) | Info (%) | True*Info (%) |
|---|---|---|---|---|---|
| 0.0 | 46.63 | 66.15 | 90.45 | 86.42 | 78.16 |
| 0.2 | 48.90 | 65.89 | 80.68 | 91.69 | 73.98 |
| 0.3 | 46.88 | 66.77 | 90.45 | 86.42 | 78.17 |
| 0.4 | 48.71 | 66.58 | 93.88 | 85.81 | 80.56 |
| 0.5 | 50.80 | 68.55 | 90.94 | 86.17 | 78.37 |
| 0.6 | 48.95 | 65.37 | 75.15 | 84.21 | 63.29 |
| 0.7 | 44.74 | 57.69 | 66.01 | 83.13 | 54.87 |
| 0.8 | 35.45 | 51.76 | 65.04 | 84.35 | 54.86 |
| 0.9 | 35.21 | 51.64 | 65.04 | 84.35 | 54.86 |

# K Limitations

First, our approach is flexible and could be used for any sort of steering, including less-noble purposes (e.g., jailbreaking, toxic content generation), which may carry negative impacts. Second, our approach is hyperparameter-dependent, and the research mainly focuses on QA tasks along specific behaviors (e.g., truthfulness and informativeness) on English datasets. Finally, since truthfulness and informativeness content lack

ground-truth for direct evaluation, we follow prior work (Li et al., 2023; Wang et al., 2024c) and employ LLM-based judge models as evaluators. It is worth noting that this evaluation is not perfect and may introduce errors.

