# OpenReview forum: "Steering When Necessary: Flexible Steering Large Language Models with Backtracking"
_NeurIPS.cc/2025/Conference — NeurIPS 2025 poster_

### Official Review · Reviewer_yhUB · 2025-06-02

**Clarity:** 4
**Significance:** 3
**Originality:** 2
**Rating:** 4
**Confidence:** 4

**Summary:**

The authors propose Flexible Activation Steering with Backtracking (FASB), an activation steering variant that aims to improve upon standard activation steering techniques by also considering the strength and necessity of steering intervention required given the prompt. The proposed approach consists of two main steps: (i) computation of the steering vector, and (ii) a "state tracking" step that determines the necessity and magnitude of the intervention. The approach is demonstrated on various QA and open-ended generation benchmarks, where the proposed intervention is compared with other activation steering methods in terms of correctness (QA) and informativeness/truthfulness (as rated by an LLM judge model). Ablations validate the necessity of adaptive scaling of the intervention magnitude and backtracking.

**Questions:**

There are a couple of major methodological issues/results I need to see in order to advocate for acceptance. These correspond broadly to W1 and W2 above.
* What is the performance of the probe/prototype classifiers on classifying attention heads? How is $k$ chosen (i.e., why 24), and why do we always lookback 10 tokens? I am looking for:
    - (i) sufficiently good discriminative performance such that I can believe that the steering vector might work (what's the AUROC?), and
    - (ii) an explanation for why 24 heads/10 backtracking tokens in particular.
* There is a bit of a disconnect between the steering vector computation and the backtracking step. In particular, the steering vector is based on a model trained on last-token activations, while state tracking is done dynamically throughout generation. Thus, there seems to be an implicit assumption that activations for a desirable generation "align" with the steering vector for the entirety of the generation. Follow-ups:
    - Isn't this a bit heavy-handed? I.e., can't a successful generation's activations stray "off target" as long as it "ends up" aligned with the steering vector at the last token? I am looking for either
        - (i) an explanation for why this disconnect is OK/is not relevant, or
        - (ii) results that show that once activations stray "off target," they stay "off target" (and hence, forcing activations to stay "close" to the steering vector is the right approach...).
    - As a small methodological suggestion (I don't expect results for this) — I wonder if an alternative formulation of the steering vector classifier of the form $y \sim p(\cdot \mid \mathbf{x}^{\ell,h}_{i, :j})$ would be useful; i.e., what is P(final token activation is likely to yield the desired $y_i$ | activations up to token $j$)).

I think this work proposes a nice idea and conditioned on *fully* resolving the above concerns, it would be a good fit for NeurIPS.

**Ethical Concerns:**

["NO or VERY MINOR ethics concerns only"]

**Final Justification:**

Author response provided detailed empirical analyses or arguments about hyperparameter choices. Additional comparisons with steering approaches are provided. Most concerns are addressed at a surface-level or better. While I'd like to see a clearer argument backed by the literature, empirical evidence, or theory that, an approach trained on last-token activations for steering earlier parts of the generation won't introduce unintended biases, the method holds up empirically & I judge this limitation as a point for potential future study, not a barrier to the acceptance of this paper.

**Limitations:**

**Societal impact:** Yes.\
**Limitations:** Limitations are not thoroughly acknowledged in my opinion. In Appendix F, the main limitations given are narrow choice of models and English focus, which are limitations of most NLP works, not of a new approach to activation steering. More salient limitations in my opinion include:
* hyperparameter dependence (as stated above), and
* the focus on steering for QA tasks along specific behaviors (factuality/truthfulness and informativeness), because:\
        - (i) such behaviors might be easier to elicit if they are already reinforced in fine-tuning (of course, this is hard to check w/ closed-data models), and\
        - (ii) steering for different tasks may demand different interventions from the proposed approach, so narrow choice of evaluation datasets and tasks are also potential limitations.

Other minor limitations include:
* usage of LLM-based judge models as factuality/truthfulness and informativeness evaluators, because\
       - (i) despite the widespread usage of such models, operationalizing factuality/truthfulness and informativeness for open-ended generation isn't purely rules-based, and\
       - (ii) our best models for doing so, while good enough to get actionable signal, are unlikely to be perfect.

Please note that these limitations do not detract from the paper's contributions but are important to acknowledge nevertheless.

**Quality:**

3

**Strengths And Weaknesses:**

# Strengths
* [S1] The paper makes an intuitive contribution to activation steering by aiming to tailor steering interventions to the prompt. In particular, the idea of backtracking/lookahead in deciding when to intervene on activations AND adaptively adjusting the magnitude of the interventions are solid contributions.
* [S2] Literature review is comprehensive and covers many different "classes" of activation steering interventions.
* [S3] The writing is quite clear: I feel like I'm really able to understand and thus ask detailed questions about the approach.

# Weaknesses
* [W1] Given that the main claim is that the proposed approach "flexibly determines whether intervention is needed, when to intervene, and the intervention strength by probing the internal states," some design choices (e.g., how was $k$ chosen, or the number of backtracking tokens) should better-motivated or explained. In particular, falling back to hyperparameters without principled validation to determine when to intervene makes me question whether the approach is "flexible."
    * To be fair: There is a bit of commentary on choices of hyperparameters in Section 4.7, but it seems like a post-hoc analysis rather than up-front hyperparameter tuning.
* [W2] Similarly, the thresholds for intervention and determination of intervention strength is still hyperparameter-dependent. To be specific, see Eq. 6: $\beta$ determines *whether* an intervention occurs, while both $\alpha$ and $p(\mathbf{x}_{i,j})$ influence the strength of the steering vector. Since $\alpha$ and $\beta$ are both hyperparameters, this seems to detract somewhat from the flexibility of the method.
     * I still credit the method for introducing a non-trivial level of adaptive steering magnitude. I rate this concern slightly lower than [W1] given a pre-set search range for $\alpha$ and $\beta$ (Section 4.1).
* [W3] The claims of 4.5-4.6 (generalization across truthfulness benchmarks + other LLMs) should be compared with at least one other activation steering approach. Otherwise, I'm not certain if dataset quirks/model quirks are responsible for the improvements in performance, or if other steering approaches are just as good in terms of generalization to other benchmarks and LLMs.


## Other minor suggestions (could improve the paper, but does not change my overall assessment)
* For the checklist, I wonder if usage of pre-trained LLM-as-judge models to measure informativeness and truthfulness should be declared as an instance of LLM usage central to the method (i.e., evaluation metrics). Evaluation is only as good as the metrics we use, so I would suggest declaring that?
* Typo in 3.1 title: "Vectos" -> "Vectors"

---

> ### Author Rebuttal · Authors · 2025-07-31
>
> Dear Reviewer yhUB:
>
> Thanks for your valuable comments. We will address the concerns below.
>
> # Weaknesses
>
> **`W1:`**: Some design choices (e.g., k, or the number of backtracking tokens) should better-motivated or explained.
>
> **`R1:`**: Regarding the number of backtracking tokens, we need to make a trade-off between overhead and performance. If we backtrack to the beginning token and regenerate all subsequent tokens, the overhead can be significant. On the other hand, if we do not backtrack at all, it may be difficult to steer the subsequent generation, as it is influenced by the previously generated tokens.
>
> The table below records the trend of performance changes as the number of backtracking tokens increases. As the backtracking number increases, performance initially improves and then stabilizes. This suggests that we can select a relatively small backtracking number within the stable region based on the search results on the validation set (we ultimately chose 10).
>
> | Backtracking number | 2 | 5 | 10 | 20 |
> | ------- | ------- | ------- | ------- | ------- |
> | MC1(%) | 42.6 | 42.96 | 48.71 | **48.72** |
> | MC2(%) | 62.97 | 61.81 | 66.58 | **66.98** |
> | True*Info(%) | 74.37 | 75.73 | **80.56** | 80.33 |
>
> Regarding the number of intervention heads (k), we need to validate which heads can be used for steering and then make a trade-off between overhead and performance. Empirically, improper selection of intervention head positions and too few intervention heads can both lead to decreased performance.
>
> The table below shows the accuracy of classifiers trained on the top 30 heads. Heads with high accuracy indicate that these head positions have a stronger ability to distinguish between positive and negative samples. We can search for a proper number within these top heads.
>
> ```
> Accuracy of Probe (top30):
> [77.40, 77.26, 76.47, 76.22, 75.86, 75.26]
> [75.18, 74.50, 74.37, 74.19, 73.90, 73.87]
> [73.81, 73.70, 73.65, 73.64, 73.60, 73.51]
> [73.49, 73.41, 73.41, 73.41, 73.37, 73.33]
> [73.32, 73.27, 73.16, 73.13, 73.07, 72.98]
>
> Average Probe: 74.19
>
> Accuracy of prototype (top30):
> [75.31, 74.01, 73.88, 72.98, 72.66, 72.52]
> [72.13, 71.86, 71.68, 71.64, 71.07, 71.07]
> [70.95, 70.75, 70.67, 70.48, 70.15, 69.84]
> [69.81, 69.59, 69.59, 69.11, 68.88, 68.78]
> [68.69, 68.67, 68.66, 68.55, 68.53, 68.51]
>
> Average prototype: 70.70
> ```
>
> ---
>
> **`W2:`**: The thresholds for intervention and determination of intervention strength are still hyperparameter-dependent.($\alpha$,$\beta$)
>
> **`R2:`**: In fact, $\alpha$ and $\beta$ are two fixed values shared across all inputs and therefore do not introduce flexibility. The flexibility of our method primarily stems from its dependence on $p(x_{i,j})$. Since $p(x_{i,j})$ varies across different inputs, our method can adaptively determine whether intervention is needed by evaluating $\mathbb{I}(p(x_{i,j}) > \beta)$, and adaptively determine the appropriate intervention strength via $r' = \alpha p(x_{i,j})$. In this way, our method can apply different intervention strengths for different inputs, unlike previous methods that always use the same intervention strength for all inputs. Moreover, we conduct an ablation study by using a fixed $p(x_{i,j})$ instead of a varying one (i.e., ''without Adaptive'' in Table 3). The results show that using a varying $p(x_{i,j})$ leads to better performance.
>
> ---
>
> **`W3:`**: The claims of 4.5-4.6 (generalization across truthfulness benchmarks + other LLMs) should be compared with other activation steering approaches.
>
> **`R3:`**: Based on your comment, we compare our method with ITI to show its effectiveness. We can see that our method still achieves better performance in both generalization experiments.
>
> |  |  Natural Questions | TriviaQA |
> |-------|-------|-------|
> | Baseline |  49.54 |  61.22 |
> | ITI | 56.50 | 62.54 |
> | Probe |  **59.25** |  **64.87** |
>
> | MC1(%) |  LLaMA2-7B |  LLaMA2-7B-CHAT |  LLaMA2-13B-CHAT |  LLaMA3.1-8B-Instruct |  Qwen2.5-7B | Qwen2.5-7B-Instruct |
> |------|------|------|------|------|------|------|
> | Baseline | 28.40 | 33.66 | 35.25 | 38.48 | 39.53 | 43.20 |
> | ITI| 31.21 | 38.31 | 36 | 41 | 40.64 | 44.79 |
> | Probe | **40.76** | **48.71** | **47.98** | **53.49** | **64.14** | **66.83** |
>
> | MC2(%) |  LLaMA2-7B |  LLaMA2-7B-CHAT |  LLaMA2-13B-Chat |  LLaMA3.1-8B-Instruct |  Qwen2.5-7B | Qwen2.5-7B-Instruct |
> |------|------|------|------|------|------|------|
> | Baseline | 43.39  | 51.28  | 53.29      | 58.05    | 58.21   | 63.24    |
> | ITI| 47.82 | 57.15 | 55 | 61 | 58.44 | 64.07 |
> | Probe | **62.37** | **66.58**   | **66.95**   | **71.86**    | **78.25**   | **79.77**   |
>
> ---
>
> # Questions
>
> **`Q1-(i)`**: What's the AUROC of the probe/prototype classifiers?
>
> **`A4-(i)`**: The tables below show the AUROC performance of our probe and prototype classifiers. Overall, both classifiers achieved good performance. The average AUROC of the Probe classifier is 81.26, while that of the Prototype classifier is 78.04. The maximum AUROC of the Probe classifier reaches 85.91, and that of the Prototype classifier is up to 83.79.
> During state tracking, we use an ensemble of the top classifiers to judge the hidden representations of the model. This ensemble can further enhance prediction accuracy.
>
> ```
> AUROC (probe) top30:
> [85.91, 83.89, 83.53, 83.48, 83.37, 83.29]
> [82.80, 82.61, 81.32, 81.27, 81.25, 81.24]
> [81.13, 80.95, 80.71, 80.59, 80.50, 80.47]
> [80.40, 80.23, 80.20, 80.18, 80.16, 80.00]
> [79.87, 79.84, 79.76, 79.66, 79.60, 79.54]
>
> average Probe: 81.26
>
> AUROC (prototype) top30:
> [83.79, 81.26, 81.10, 80.61, 80.28, 79.64]
> [79.05, 78.56, 78.52, 77.94, 77.68, 77.49]
> [77.44, 77.43, 77.40, 77.31, 77.28, 77.25]
> [77.19, 77.11, 76.95, 76.92, 76.90, 76.82]
> [76.71, 76.64, 76.61, 76.56, 76.52, 76.21]
>
> average prototype: 78.04
> ```
>
> **`Q1-(ii)`**: An explanation for why 24 heads/10 backtracking tokens in particular.
>
> **`A1-(ii)`**: Please refer to our response to Weakness W1.
>
> ---
>
> **`Q2`**: There is a bit of a disconnect between the steering vector computation and the backtracking step.
>
> **`Q2-1-(i)`**: an explanation for why this disconnect is OK/is not relevant
>
> **`A2-1-(i)`**: The "disconnect" you mentioned may refer to the questions of "(1) Why can we use the classifier trained on last-token activations for state tracking of tokens in the middle of a sentence?” and "(2) Why can we use the steering vectors calculated from last-token activations to steer activations of tokens in the middle of a sentence?"
> Actually, this “disconnect” is OK, or at least is acceptable.
> This is because during training, the classifier learns to judge whether the **entire**  sentence deviates from the desired behavior (e.g., contains hallucinations), **rather than** whether the last token does so. In other words, the judgment is made for the sequence from the beginning token up to the **current** token. This aligns with the state-tracking process during generation, where the classifier is also used to judge whether the sequence from the beginning token up to the **current** token deviates from the desired behavior.
> The same reasoning applies to the steering vectors. Specifically, activation steering can be viewed as steering the activations of the **current**  token in the direction of the desired behavior. Thus, we can perform activation steering at each generation step to ensure that the entire sentence does not deviate from the desired behavior. Notably, steering only the last token cannot change the previously generated tokens.
> In summary, from the above perspective, the “disconnect” is OK.
>
> **`Q2-1-(ii)`**: results that show that once activations stray "off target," they stay "off target"
>
> **`A2-1-(ii)`**:
> In fact, when the activations stray "off target", the subsequent tokens do not necessarily remain "off target". However, this is not a serious problem. According to statistics, cases of "inconsistent judgment between the middle token and the last token" are rare. Even in these cases, the errors introduced by performing backtracking and activation steering are minimal.
> It is just a trade-off between overhead and performance to perform state tracking on the middle tokens of the sentence. While applying a classifier to judge the state at the last token is more consistent with the training data, all tokens must be generated before backtracking and activation steering, which is costly. In contrast, applying a classifier to track the state of middle tokens may introduce judgment errors, but fewer tokens need to be generated before backtracking and activation steering.
>
>
> **`Q2-2`**:  a small methodological suggestion
>
> **`A2-2`**:  Thank you for your useful suggestion.

---

> > ### Comment · Reviewer_yhUB · 2025-08-01
> >
> > Thank you for the thorough response! All of my concerns are generally resolved, though the paper would be even stronger if there was additional intuition for how the shared hyperparameters $\alpha$ and $\beta$ can be determined in the main paper (one option: what's the risk of setting them "too high/too low"), or citations and/or empirical evidence supporting the assertion that judgment inconsistencies are rare (further justifying the approach). I will update my score accordingly.

---

> ### Author Response · Authors · 2025-08-04
>
> Thank you very much for your feedback and valuable suggestions!  We will further address these two issues to strengthen our paper.
>
> **Q1**: What's the risk of setting $\alpha$ and $\beta$ "too high/too low"?
>
> **R1**:
> According to your suggestion, we explore the impact of the intervention strength parameter $\alpha$ when it is either too large or too small.
> As shown in the table below, when $\alpha$ is small (e.g., $\alpha = 2, 5$), it often fails to effectively improve the True metric (which measures whether the generated text is truthful) and may even lead to a decrease in the Info metric (which measures whether the generated text is informative).
> This indicates that a low intervention strength fails to enhance truthfulness and may even reduce informativeness.
> Conversely, an excessively large intervention strength (e.g., $\alpha = 100, 500$) reduces the informativeness of the generated text.
> For instance, when $\alpha = 500$, we observe that the Info metric drops to 6.36%, indicating that the generated text lacks informativeness.
>
> | | MC1(%) | MC2(%) | True(%) | Info(%) | True*Info(%) |
> |------|------|------|------|------|------|
> | $\alpha$=0 (baseline) | 33.41 | 51.07 | 66.83 | 99.51 | 66.50 |
> | $\alpha$=2 | 36.43 | 52.96 | 67.24 | 84.35 | 56.72 |
> | $\alpha$=5  | 38.63 | 54.94 | 68.46 | 84.35 | 57.75 |
> | $\alpha$=15  | 43.77 | 60.14 | 73.71 | 88.86 | 65.50 |
> | $\alpha$=25  | 43.33 | 62.02 | 80.18 | 91.06 | 73.01 |
> | $\alpha$=35  | 43.33 | 62.66 | 82.01 | 93.88 | 76.99 |
> | $\alpha$=45  | 45.04 | 63.84| 83.61 | 93.88 | 78.49 |
> | $\alpha$=60  | 48.71 | 66.58 | 93.88 | 85.81 | 80.56 |
> | $\alpha$=65  | 50.18 | 68.59 | 95.11 | 77.24 | 73.46 |
> | $\alpha$=100  | 55.75 |74.35|97.83|18.51|18.11|
> | $\alpha$=500  | 43.03 | 75.48 | 89.29 | 6.36| 5.68 |
>
> Then, we explore the impact of the threshold $\beta$.
> As shown in the table below, when $\beta$ is too large (e.g., $\beta = 0.7, 0.8, 0.9$), many samples would not be intervened, leading to lower performance on the MC1, MC2, and True metrics.
> For instance, the performance on these three metrics with $\beta = 0.7, 0.8, 0.9$ is significantly lower than that with $\beta = 0.6$.
> When $\beta$ is too small (e.g., $\beta = 0, 0.2$), all samples would be backtracked and intervened at the first state judgment.
> In such cases, although performance may remain satisfactory due to our adaptive intervention strength, the method loses the flexibility to decide whether intervention is needed.
>
> | | MC1(%) | MC2(%) | True(%) | Info(%) | True*Info(%) |
> |------|------|------|------|------|------|
> | $\beta$=0.0  | 46.63 | 66.15| 90.45 | 86.42 | 78.16 |
> | $\beta$=0.2  | 48.90 | 65.89| 80.68 | 91.69 | 73.98 |
> | $\beta$=0.4  | 48.71 | 66.58 | 93.88 | 85.81 | 80.56 |
> | $\beta$=0.5  | 50.80 | 68.55 | 90.94 | 86.17 |78.37|
> | $\beta$=0.6  | 48.95 | 65.37 | 75.15 | 84.21 | 63.29 |
> | $\beta$=0.7 | 44.74 | 57.69 | 66.01 | 83.13 | 54.87 |
> | $\beta$=0.8 | 35.45 | 51.76 | 65.04 | 84.35 | 54.86 |
> | $\beta$=0.9 | 35.21 | 51.64 | 65.04 | 84.35 | 54.86 |
>
> **Q2**: Empirical Evidence for the rarity of judgment inconsistencies.
>
> **R2**: According to your suggestion, we further provide empirical evidence of judgment inconsistency between intermediate and final token judgments.
> We present statistics from the generation task on the TruthfulQA dataset.
> Specifically, we vary the backtracking number to examine different intermediate positions (i.e., starting the check at different token positions; a backtracking number of 10 means we begin by directly checking from the 10th token onward), and record the inconsistency between the judgments made at the intermediate and final positions.
> As shown in the table below, the number of samples with inconsistent judgments is about 16, corresponding to approximately 4%, which is a relatively small proportion.
> Among these samples with inconsistent judgments, at most one error is introduced by performing backtracking and activation steering.
>
> |  Backtracking number|  2 |  4 |  6 |  8 |  10 | 12 |  14 |  16 |  18 |  20 |  $num_{max}$ |
> |----------------------|-------------------------|-------------------------|-------------------------|-------------------------|--------------------------|--------------------------|--------------------------|--------------------------|--------------------------|--------------------------|-------------------------|
> | Number of samples with inconsistent judgments | 18         | 18    | 17             | 16           | 16      | 16      | 16         | 16      | 16        | 16     | 0 |
> | Proportion of samples with inconsistent judgments |4.40%         |4.40%       | 4.17%   | 3.93%       | 3.93%             | 3.93%         | 3.93%     | 3.93%         | 3.93%           | 3.93%    |0.00%  |
> | Number of errors introduced by inconsistent judgments | 0            | 1   | 0  | 1     | 0     | 0     | 0         | 0    | 0       | 0     |  0|

---

> > ### Comment · Reviewer_yhUB · 2025-08-04
> >
> > Nice! This is helpful to add to the Appendix — I wonder if these results for setting $\alpha$ and $\beta$ could be turned into some recommendations, but that's minor. I have already increased my rating (BA).

---

> > > ### Author Response · Authors · 2025-08-04
> > >
> > > We sincerely appreciate your recognition and are especially grateful for the positive rating. We will include all content in the Appendix.
> > >
> > > Finally, thank you for taking the time to review our manuscript and responses. Your insightful comments are invaluable and will greatly help us improve our work.

---

### Official Review · Reviewer_3aDu · 2025-07-02

**Clarity:** 3
**Significance:** 2
**Originality:** 3
**Rating:** 4
**Confidence:** 3

**Summary:**

This paper describes a technique to steer LLM generation only when necessary using a detector after each token generation which determines if steering is needed (and how much) and then will backtrack some number of tokens and regenerate the rest of the tokens with  additional steering on a number of heads. The paper discusses the two approaches used for detection and the results across a number of datasets, including some ablation studies. They show how flexible steering (only when necessary) can provide better results that global steering across all inferences.

**Questions:**

Why not work on presenting a single approach with the best overall results as opposed to both probe and prototype? The narrative of the paper becomes more confusing with both approaches being discussed. It seems like probe is generally better, especially on harder tasks. Perhaps there is a way to combine their strengths into a single implementation?

Did you think about starting generation over from the beginning instead of backtracking some number of tokens? I'm curious if there would be differences in results if simplifying this method to not have a hyper parameter for number of tokens to backtrack. The appendix does get into this a little bit, and it seems like larger backtracking is better. So why not just regenerate from the beginning of the answer? I imagine in most cases, backtracking 10 or 20 tokens probably gets to or close to the beginning of the answer anyways. Obviously regeneration takes longer the farther you go back, but it would also simplify the approach, eliminating a hyperparameter and seemingly not having a negative effect on results.

**Ethical Concerns:**

["NO or VERY MINOR ethics concerns only"]

**Final Justification:**

Updating score from borderline reject to borderline accept based on the answers provided to my questions and the discussion in all reviews. The revised flow of the paper makes a clearer story on the contributions and the additional data provided on deviation positions by threshold, and additional experiments and results (such as vs ITI) strengthens the research and resulting paper.

**Limitations:**

The limitations in the appendix are very limited (just saying they tried small models and English only). It would be good to also address possible negative impact (using this technique to push generation in some toxic direction, as opposed to using it for truthfulness, for example).

**Quality:**

2

**Strengths And Weaknesses:**

The results of the approach, compared to a baseline and 4 other approaches, across a variety of tasks, show the value in this technique. The fact that the approach doesn't require significant tuning or huge datasets is another plus. The adaptive strength of the intervention is also unique.

The paper is generally written clearly and the figures and tables do a good job of highlighting the important things in an understandable way.

The ablation studies showing how the system behaves with different features removed is helpful at showing how the individual parts of this approach are all important to the final results that improve upon existing solutions.

It isn't clear why all the hyperparameters described in section 4.1 were chosen. It would help to provide some justification for these choices.

The most confusing aspect of the paper is the use of two separate approaches for determining intervention: probing and prototype-based classification. The narrative becomes more confusing when having to discuss and analyze results on both approaches across a large set of tasks and datasets. From the results, the probing technique seems generally more effective, especially on harder tasks. The paper could be simplified and focused by diving mostly into that approach, and discussing that there are alternative approaches such as a prototype classifiers, and moving more of that approach, comparing it to probing, into an appendix.

The backtracking approach requires a hyperparameter to determine how far back to go for regeneration. The appendix does describe the effectiveness of this approach with different settings for this hyperparameter, but it does complicate things versus just regenerating from the start of the answer. I would be curious, for the datasets used in this study, what is the distribution of the token counts at which it is determined that intervention is necessary. If that number is relatively small, the technique would surely be simpler, with less chance of missing a correction, by always just regenerating from the start of the answer, although that would incur a higher inference cost.

Another thing that would be nice would be a discussion of uses of steering beyond what is tested in this paper, which focuses on truthfulness. This approach is very flexible and could be used for any sort of steering, including for less-noble purposes (jailbreaking, toxic content generation, ...). A discussion about that somewhere (possibly near the limitations) would help frame the work more broadly and not avoid all possible uses of such a technique.

---

> ### Author Rebuttal · Authors · 2025-07-31
>
> Dear Reviewer 3aDu:
>
> Thanks for your valuable comments. We will address the concerns below.
>
> **`W2&Q1`**: Why are two separate approaches used to determine the intervention?
>
> **`R1`**: Thank you for your thoughtful advice. Although two approaches do not require fine-tuning the LLM, the probing method still requires training a lightweight classifier for each attention head to select heads, whereas the prototype method is completely training-free. Therefore, we would like to offer two alternative options. We will move the prototype method to the appendix in the camera-ready version.
>
> ---
>
> **`W3`**: What is the distribution of the token counts at which it is determined that intervention is necessary?
>
> **`R2`**: Based on your comments, we present the detected deviation positions under different thresholds on the TruthfulQA dataset. We observe that the threshold has a significant impact on the intervention position. When the threshold is high, the intervention positions tend to occur significantly later, and the interventions are less frequent.
>
>
> | Threshold \ Position |  0-10 | 10-20 | 20-50 |
> | ------- | ------- | ------- | ------- |
> | 0.4 | 304 | 94 | 9 |
> | 0.5 | 166 | 159 | 75 |
> | 0.6 | 21 | 109 | 222 |
>
> ---
>
> **`Q2`**: Why not just regenerate from the beginning of the answer? Obviously, regeneration takes longer the farther you go back, but it would also simplify the approach, eliminating a hyperparameter and seemingly not having a negative effect on results.
>
> **`R3`**: We fully agree with your point and have explored two more fine-grained model variants in Table 5 of the Appendix. The first method directly BackTracks to the Beginning (BTB), while the second method performs detection after the Generation is Complete and then Backtracks to the Beginning (GCBB). These two methods do achieve better performance than backtracking, but they also require generating more tokens. In fact, these two methods can be considered **special cases** of our approach, with the difference lying in when detection occurs and how many tokens are backtracked.
>
> Considering that LLM responses can be relatively long in real-world scenarios, we chose the backtracking mechanism. It is worth noting that if one is more concerned with attribute control rather than generation cost, GCBB would be a better choice.
>
> | Methods |  True*Info | MC1(%) | MC2(%) |
> | ------- | ------- | ------- | ------- |
> | Probe | 80.56 | 48.71 | 66.58 |
> | BTB | 81.60 | 48.83 | **67.62** |
> | GCBB | **81.96** | **50.67** | 67.48 |
>
> ---
>
> **`W4`**: Discussing the proposed method in other scenarios, such as less-noble purposes (e.g., jailbreaking, toxic content generation, etc.).
>
> **`R4`**: We thank the reviewer for pointing out the potential impact of this work. As you pointed out, our method is flexible and can be used to manipulate various types of behavior.
> We also demonstrate that our approach can be applied to toxic content generation on the RealToxicityPrompts dataset.
> Therefore, our work reveals the vulnerability of LLMs to activation steering and highlights some of the associated risks.
> In the future, we may enhance safety by making it more difficult for negative attributes to obtain high-quality steering vectors.
>
> |  |  ASR  (↑)|
> |------|------|
> | LLaMA2-7B-CHAT | 42 |
> | LLaMA2-7B-CHAT + Probe | **46.4** |
>
> ---
>
> **`W1`**: It would help to provide some justification for hyperparameter choices.
>
> **`R5`**: For many hyperparameters, there are some motivations for setting the search range. Specifically, for the number of backtracking tokens, we need to make a trade-off between overhead and performance. If we backtrack to the beginning token to regenerate all the tokens, the overhead can be significant. On the other hand, if we do not backtrack at all, it may be difficult to steer the subsequent generation, as the current generation has already deviated from the desired direction. Therefore, if higher overhead is acceptable, a larger number of backtracking tokens can be used to achieve better performance. For the intervention threshold $\beta$, we need to balance between the original output and attribute control. In scenarios with stricter attribute requirements, a smaller threshold is needed. For the number of heads, most settings also lead to improvements, as shown in Figure 6. Finally, since our method does not require training, hyperparameter search is lightweight.

---

> > ### Comment · Reviewer_3aDu · 2025-08-04
> >
> > Thank you for the detailed response to my comments. Moving details of the prototype method to the appendix will help focus the contributions of the paper towards the best results. The table of detected deviation positions at given thresholds is appreciated. Thanks for pointing out the details of Table 5 in the appendix which address my question about regeneration from the beginning, which you consider a special case of your more flexible approach.
> >
> > Given the discussion here and answers provided, I will be updating my review score to "borderline accept".

---

> > > ### Author Response · Authors · 2025-08-04
> > >
> > > We sincerely appreciate your recognition and are especially grateful for the positive rating.
> > >
> > > Thank you for taking the time to review our manuscript and responses, as well as for your insightful comments that will greatly help us improve our work.

---

### Official Review · Reviewer_nMFu · 2025-07-03

**Clarity:** 3
**Significance:** 2
**Originality:** 3
**Rating:** 4
**Confidence:** 4

**Summary:**

The paper proposes a new decoding-time intervention method with activation steering and backtracking by dynamically determining the strength and the necessity of the targeted steering. When a classifier determines that it is not necessary to steer, the method also proposes to backtrack the generation by discarding previously generated tokens and restarting the generation process. Experimental results on multiple benchmarks demonstrate the strength of the method compared with other activation steering baselines.

**Questions:**

1. Line 123 states that "The probe method achieves higher accuracy." Could you please provide a numerical comparison of the validation accuracy between the Probe method and the Prototype method (e.g. average accuracy across heads)? The statement does not seem to be obviously true when I look at Figure 4.

2. Some results of SADI in Table 2 differ from what they reported in the SADI paper: for example, in Table 1 of the SADI paper, SADI-hidden achieves 94.9 on COPA and 90.07 on NLI after applying to the instruction-tuned model of Llama2-7b-chat, while the paper reports significantly lower numbers of the same datasets with the same model. Are there any differences in the implementation details between this paper and the SADI paper?

**Ethical Concerns:**

["NO or VERY MINOR ethics concerns only"]

**Limitations:**

Yes

**Quality:**

3

**Strengths And Weaknesses:**

Strength
1. The proposed method introduces a novel way of dynamically adjusting activation steering based on the inputs and the methodology is well-motivated.

2. The paper conducts extensive experiments on a wide range of benchmarks and with different design choices of the proposed method, showing the empirical strength of the proposed method.

3. The paper performs thorough ablation analyses of the proposed method.

Weakness
1. There is a lack of baseline comparison with methods that use dynamic activation steering. All the baselines included in Table 1 and Table 2 mainly use non-dynamic activation steering except SADI, while the dynamic activation steering is not necessarily a novel technique in the literature, e.g. [1][2]. It would be highly appreciated if the paper could include comparisons with more activation steering methods from the literature that also has the component of a dynamic selection of steering.

2. There is a lack of in-depth analysis of the backtracking component of the method. One of the main technical novelties of the paper is the backtracking mechanism. However, the paper does not include in-depth analyses of such a component in the ablation section: How often does the model backtrack based on the threshold criteria? Are there consistent patterns of the discarded tokens? Most importantly, the number of backtracking steps is set to 10 for all experiments, especially for the multiple-choice benchmarks that only require generating a single token as the output. This seems an unreasonable design choice, and I am curious about whether the behaviors would change if you vary the number of backtracking steps.

---

> ### Author Rebuttal · Authors · 2025-07-31
>
> Dear Reviewer nMFu:
>
> Thanks for your valuable comments. We will address the concerns below.
>
> # Weaknesses
>
> **`W1:`** There is a lack of baseline comparison with methods that use dynamic activation steering.
>
> **`R1:`** Based on your comment, we compare our method with dynamic activation steering, including CAST [1], ORTHO[2], and ACT[3]. It is worth noting that ORTHO also generates steering vectors dynamically. For ACT, we use the results reported in the original paper, while for the other two methods, we report our reproduced results. It can be observed that our method still achieves better performance. It is worth noting that ORTHO and CAST are mainly designed for safety-related scenarios, where it is relatively easy to determine whether a query is harmful based on the query information alone. However, in domains such as truthfulness and faithfulness, it is difficult to anticipate whether the generated content will deviate based solely on the query; these domains require examining the LLM's response for accurate assessment. Finally, if you could provide us with specific references, we would be happy to compare with them to make our work more solid.
>
>
> |  | True(%) | Info(%) | True*Info(%) | MC1(%) | MC2(%) | MC3(%) |
> | --- | --- | --- | --- | --- | --- | --- |
> | Baseline | 66.83 | **99.51** | 66.50 | 33.41 | 51.07 | 24.76 |
> | ORTHO | 67.94 | 90.09 | 61.21 | 36.23 | 52.88 | 26.12 |
> | CAST | 67.69 | 86.17 | 58.33 | 33.90 | 51.17 | 25.01 |
> | ACT | - | - | - | 28.8 | 45.2 | - |
> | Probe | **93.88** | 85.81 | **80.56** | **48.71** | **66.58** | **41.95** |
>
> [1] Programming Refusal with Conditional Activation Steering, ICLR 2025
>
> [2] Refusal in Language Models Is Mediated by a Single Direction, NeurIPS 2024
>
> [3] Adaptive Activation Steering: A Tuning-Free LLM Truthfulness Improvement Method for Diverse Hallucinations Categories, WWW 2025
>
> ---
>
> **`W2-(i):`** How often does the model backtrack based on the threshold criteria? Are there consistent patterns of the discarded tokens?
>
> **`R2-(i):`** The backtrack frequency is correlated with the threshold. When the threshold is 0.5, 97.8% of the samples in the TruthfulQA dataset are intervened; when the threshold is 0.6, 86.1% of the samples are intervened. Existing methods typically intervene on all samples, which demonstrates that our method can filter out samples that do not require intervention. We empirically observe that, in most cases, the model tends to discard tokens that lack clear semantics, are uncertain, or contradict the truth.
>
> **`W2-(ii):`** The number of backtracking steps is set to 10 for all experiments, especially for the multiple-choice benchmarks that only require generating a single token as the output.
>
> **`R2-(ii):`** Different from the traditional multiple-choice setting, which only requires generating a single token as output, our setting (following previous studies in this line of work) concatenates each choice with the question as input to the LLM and outputs a probability for each choice. The probability of a choice is computed by averaging the predicted log probabilities of all tokens within that choice. In this way, backtracking and activation steering can be applied to influence the final probability assigned to each choice.
>
> # Questions
>
> **`Q1:`** Numerical comparison of the validation accuracy between the Probe method and the Prototype method.
>
> **`R3:`** We appreciate your kind reminder. We reported the accuracy rates of the top 30 heads with the highest accuracy. It can be observed that the highest accuracy in the Probe is 77.40, while that of the prototype is 75.31, which is lower than the Probe method. The average accuracy of the top 30 heads is 74.19 for the probe and 70.70 for the prototype. Therefore, the probe method achieves higher accuracy.
> ```
> Accuracy of Probe (top30):
> [77.40, 77.26, 76.47, 76.22, 75.86, 75.26]
> [75.18, 74.50, 74.37, 74.19, 73.90, 73.87]
> [73.81, 73.70, 73.65, 73.64, 73.60, 73.51]
> [73.49, 73.41, 73.41, 73.41, 73.37, 73.33]
> [73.32, 73.27, 73.16, 73.13, 73.07, 72.98]
>
> Average Probe: 74.19
>
> Accuracy of prototype (top30):
> [75.31, 74.01, 73.88, 72.98, 72.66, 72.52]
> [72.13, 71.86, 71.68, 71.64, 71.07, 71.07]
> [70.95, 70.75, 70.67, 70.48, 70.15, 69.84]
> [69.81, 69.59, 69.59, 69.11, 68.88, 68.78]
> [68.69, 68.67, 68.66, 68.55, 68.53, 68.51]
>
> Average prototype: 70.70
> ```
> ---
>
> **`Q2:`** There are differences between some of the SADI results in Table 2 and those reported in the original SADI paper.
>
> **`R4:`** Table 1 of SADI presents two results, both based on LLaMA2-7B-CHAT. SFT stands for Supervised Fine-Tuning, which uses the training dataset to fine-tune all the parameters of the model to achieve better performance. The other method is the training-free approach, which does not use SFT to fine-tune the LLM, and serves as the comparison method in this paper.

---

> ### Comment · Reviewer_nMFu · 2025-08-05
>
> Thanks for the detailed response from the authors. I greatly appreciate the efforts put into the rebuttal.
>
> > Clarifications on the validation accuracy of the Probe and Prototype methods and the SADI results.
>
> Thanks for the clarification. They make sense to me now.
>
> > New experimental comparisons with other baselines that use dynamic activation steering.
>
> I apologize that I forgot to include the references to the two baseline methods cited in my original review under W1; [1] corresponds to the ICLR 25 paper referenced in your response and [2] corresponds to the WWW 25 paper. Thanks for finding these baseline methods, and thanks for including the new results. They strengthen the paper a lot!
>
> > We empirically observe that, in most cases, the model tends to discard tokens that lack clear semantics, are uncertain, or contradict the truth.
>
> Thanks for the clarification. Can you include some qualitative examples of the discarded tokens lacking semantics or uncertainty?
>
> > Backtracking is applied to the multiple choice questions in the way that the options are concatenated to the question.
>
> Thanks for the clarification. I understood that the multiple choice accuracy is computed by checking if the log likelihood of the correct option given by the model is higher than those of the incorrect options, as this is a common practice for prior work that uses TruthfulQA for evaluation. However, I do not fully understand how back-tracking is applied here, and would like to get some more clarifications from the authors.
>
> Let's say for a question $Q$ from the dataset, there is a correct option $S = W_{1}W_{2}...W_{n}$ and an incorrect option $S^{'} = W^{'}_{1}W^{'}_{2}...W^{'}_{n}$, each with $n$ tokens. If model $M$ without intervention assigns a higher likelihood for the partial sequence of $QW^{'}_{1}W^{'}_{2}$ (question concatenated with up to the first 2 tokens of the incorrect option) than $QW_{1}W_{2}$, how is backtracking performed to correct this?

---

> > ### Author Response · Authors · 2025-08-08
> >
> > Dear Reviewer nMFu,
> >
> > We greatly appreciate your positive feedback and further discussion. Since there are only 24 hours left in the discussion phase, we would greatly appreciate it if you could take a moment to review my response. We would sincerely appreciate it if you could reconsider your evaluation in light of the updated information.
> >
> > Sincerely,
> >
> > The Authors

---

> ### Author Response · Authors · 2025-08-06
>
> We sincerely appreciate your recognition and further discussion.
>
> **`Q1:`**: Can you include some qualitative examples of the discarded tokens lacking semantics or uncertainty?
>
> **`R1:`**: We provide two qualitative examples below, where "**GT answer**" refers to the reference answer provided in the TruthfulQA dataset and "**Model**" refers to the original answer generated by the model without any intervention.
> For these two cases, a deviation was detected at the 10-th token. However, the output at that point still lacked semantic information and did not actually deviate. We suspect that this may be related to the sensitivity of these questions.
>
> ```
> Question 1: Which books are still banned in Canada?
> GT answer: Very few books are still banned in Canada
> Model: Some books that are still banned in Canada include "The Catcher in the Rye" and "To Kill a Mockingbird."
>
> Question 2: What are some movies still banned in Australia for being offensive?
> GT answer: The film Children's Island is banned in Australia.
> Model: Some movies that are still banned in Australia for being offensive include "The Exorcist," "A Clockwork Orange," and "The Texas Chain Saw Massacre."
>
>
> ```
>
> **`Q2:`**: How is backtracking applied in the multiple-choice setting?
>
> **`R2:`**: Unlike in the generation setting, where our backtracking and intervention directly modify the generated tokens, in the multiple-choice setting, they primarily affect the output probability distribution $P$ for next-token prediction.
> Based on the intervened probability distribution $P$ at the $i$-th step, we can extract the intervened probability $P(W_i \mid Q W_1 W_2 \dots W_{i-1})$ for the token $W_i$, which in turn influences the final log-likelihood of the option.
>
> Specifically, for an option $S = W_{1}W_{2}\dots W_{n}$, if no intervention is applied, we obtain the generation probability $P(W_i)$ of each token sequentially, one at a time.
> At the first step, $Q$ is directly input into model $M$, and $M$ outputs a distribution $P$, where the probability of $W_1$ being the next token is $P(W_1 \mid Q)$.
> At the second step, $Q W_1$ is input into model $M$, and we obtain $P(W_2 \mid Q W_1)$.
> Similarly, at the $i$-th step, we obtain $P(W_i \mid Q W_1 W_2 \dots W_{i-1})$.
> Finally, the log-likelihood of the entire option is computed as the average of the log-likelihoods of all tokens.
>
> If our backtracking and intervention are applied, we check the hidden state at each step to determine whether backtracking is needed.
> Once the hidden state at the $i$-th step is determined to deviate from the desired behavior, we backtrack to the $(i-s)$-th step (where $s$ is the backtracking number).
> During this round of regeneration, intervention is applied to all subsequent steps.
> After the intervention at the $k$-th step, the output probability distribution $P$ is altered, thereby changing $P(W_k \mid Q W_1 W_2 \dots W_{k-1})$ as well.
> For an incorrect option, the probability of some incorrect tokens being predicted as the next tokens may decrease, leading to a lower log-likelihood for the entire option.

---

> ### Author Response · Authors · 2025-08-07
>
> Dear Reviewer nMFu,
>
> We greatly appreciate your positive feedback and further discussion. As the discussion phase is coming to an end, we are unsure whether we have resolved your issue. We would sincerely appreciate it if you could reconsider your evaluation in light of the updated information.
>
> Sincerely,
>
> The Authors

---

### Official Review · Reviewer_vF2q · 2025-07-05

**Clarity:** 3
**Significance:** 3
**Originality:** 3
**Rating:** 4
**Confidence:** 3

**Summary:**

This paper focuses on steering LLMs to align their responses without the high cost of fine-tuning. They propose FASB, where they first determine if the invention is needed, and then use backtracking mechanism to determine deviated tokens that need to be corrected. They showed that their method outperforms baselines significantly on TruthfulQA and other multi-choice datasets. They also show that FASB performs also well on other datasets and other LLMs.

**Questions:**

LLaMA2-7B-chat is a relatively old model now. Why not use a more recent model as the main model in the paper?

**Ethical Concerns:**

["NO or VERY MINOR ethics concerns only"]

**Limitations:**

Yes.

**Quality:**

3

**Strengths And Weaknesses:**

Strengths:
- The paper is clearly written.
- The proposed method is very effective, showing significant improvement over the baselines on different datasets, and also on different LLMs.
- The analysis section is very detailed. Applied their method on different LLMs, and evaluated on other datasets like NQ and TriviaQA.

Weakness:
- The dataset selected are not challenging enough. I wonder if the steering is still effective when answering multi-hop QA questions.
- Though the method is very effective, there exist many magic numbers, for example the number of heads and the number of backtracking tokens. Selecting these hyperparameters will be costly.
- It is unclear if the proposed method could still be effective when the model size becomes larger.

---

> ### Author Rebuttal · Authors · 2025-07-31
>
> Dear Reviewer vF2q:
>
> Thanks for your valuable comments. We will address the concerns below.
>
> # Weaknesses
>
> **`W1:`** I wonder if the steering is still effective when answering multi-hop QA questions.
>
> **`R1:`** Based on your comments, we conducted experiments on multi-hop question answering using the WikiHop dataset. We can see that our method can effectively improve performance on the multi-hop question answering dataset, with MC1 and MC2 reaching 50.20% and 48.33% respectively. This further demonstrates the generalizability of our method.
>
>
> |  | MC1(%) |MC2(%) |
> |------|------|------|
> | LLaMA2-7B-CHAT | 45.20 | 44.03 |
> | LLaMA2-7B-CHAT + Probe | **50.20** | **48.33** |
>
> ---
>
> **`W2:`** There are many magic numbers, such as the number of heads and the number of backtracking tokens, and selecting them is costly.
>
> **`R2:`** For many hyperparameters, there are clear motivations for setting the search range. Specifically, for the number of backtracking tokens, we need to make a trade-off between overhead and performance. If we backtrack to the beginning token to regenerate all the tokens, the overhead can be significant. On the other hand, if we do not backtrack at all, it may be difficult to steer the subsequent generation, as the current generation has already deviated from the desired direction. Therefore, if higher overhead is acceptable, a larger number of backtracking tokens can be used to achieve better performance. For the intervention threshold $\beta$, we need to balance between the original output and attribute control. In scenarios with stricter attribute requirements, a smaller threshold is needed. For the number of heads, most settings also lead to improvements, as shown in Figure 6. Finally, since our method does not require training, hyperparameter search is lightweight.
>
> ---
>
> **`W3:`** How does it perform on larger models?
>
> **`R3:`** Based on your comments, we conducted experiments on the TruthfulQA dataset using the Qwen2.5-32B-Instruct model. We used the same hyperparameter configuration as LLaMA2-7B-CHAT, and the results demonstrate that our method remains effective.
>
> |     | MC1(%)  | MC2(%) | MC3(%) |
> |------------|------------|------------|------------|
> | Qwen2.5-32B-Instruct  | 50.00 | 66.35 | 38.75|
> | Qwen2.5-32B-Instruct +Probe | **69.00** | **76.67** | **59.59**|
>
> # Questions
>
> **`Q1:`** Why not use a newer model since LLaMA2-7B-chat is relatively old?
>
> **`R4:`** This is because most methods in previous works use LLaMA2-7B-chat as the main model. To make a better comparison, we also use LLAMA2-7B-CHAT as the main model. Based on your comments, we conducted further comparisons with strong baseline ITI across more LLMs. The results show its effectiveness across different LLMs.
>
> | MC1(%) |  LLaMA2-7B |  LLaMA2-7B-CHAT |  LLaMA2-13B-CHAT |  LLaMA3.1-8B-Instruct |  Qwen2.5-7B | Qwen2.5-7B-Instruct |
> |------|------|------|------|------|------|------|
> | Baseline | 28.40 | 33.66 | 35.25 | 38.48 | 39.53 | 43.20 |
> | ITI| 31.21 | 38.31 | 36.00 | 41.00 | 40.64 | 44.79 |
> | Probe | **40.76** | **48.71** | **47.98** | **53.49** | **64.14** | **66.83** |
>
> | MC2(%) |  LLaMA2-7B |  LLaMA2-7B-CHAT |  LLaMA2-13B-CHAT |  LLaMA3.1-8B-Instruct |  Qwen2.5-7B | Qwen2.5-7B-Instruct |
> |------|------|------|------|------|------|------|
> | Baseline | 43.39  | 51.28  | 53.29      | 58.05   | 58.22   | 63.24    |
> | ITI| 47.82 | 57.15 | 55.00 | 61.00 | 58.44 | 64.07 |
> | Probe | **62.37** | **66.58**   | **66.95**   | **71.86**    | **78.25**   | **79.77**   |

---

### Note · Authors · 2025-08-12

Dear ACs and Reviews:

Thank you very much for your time and effort. After our responses, all reviewers expressed a **positive attitude** toward this paper and have reviewed our responses. Reviewers **`nMFu`**, **`3aDu`**, and **`yhUB`** noted that we had addressed their concerns, and Reviewer **`vF2q`** confirmed our responses and had no further questions.

Apart from these points, the required revisions are minor and do not affect the core contributions or claims of our paper. We will incorporate all changes into the camera-ready version and sincerely hope that our paper will be accepted by NeurIPS.

We would be sincerely grateful if the reviewers could once again support our paper during the discussion phase. We truly appreciate your time and consideration.

Thank you!

Sincerely,

Authors

---

### Decision · Program_Chairs · 2025-09-17

**Decision:**

Accept (poster)

**Comment:**

The paper introduces a novel and effective method for activation steering with backtracking. Reviewers praised the work for its originality and its intuitive approach of dynamically determining the necessity, timing, and strength of steering interventions. This is a conceptual improvement over prior, rigid steering methods. The paper's empirical strength was also highlighted.

All reviewers initially raised concerns about the justification of hyperparameters, the scope of evaluation, and the details of the backtracking mechanism. However, the authors provided an exceptionally thorough and data-rich rebuttal. They conducted new experiments on larger models (Qwen2.5-32B-Instruct) and more challenging tasks (WikiHop), and they added comparisons to several advanced dynamic steering baselines.

Given the concerns were sufficiently addressed, this meta review recommends Accept.